DOI: 10.1038/s41467-018-06968-7　　**OPEN**

# PRMT2 links histone H3R8 asymmetric dimethylation to oncogenic activation and tumorigenesis of glioblastoma

Feng Dong[1], Qian Li[1], Chao Yang[2,3], Dawei Huo[1], Xing Wang[1], Chunbo Ai[1], Yu Kong[1], Xiaoyu Sun[1], Wen Wang[4], Yan Zhou [4], Xing Liu[5], Wei Li[6], Weiwei Gao[7], Wen Liu[7], Chunsheng Kang[2,3] & Xudong Wu[1,2]

Transcriptional deregulation has a vital role in glioblastoma multiforme (GBM). Thus, identification of epigenetic modifiers essential for oncogenic transcriptional programs is a key to designing effective therapeutics for this deadly disease. Here we report that Protein Arginine Methyltransferase 2 (PRMT2) is highly expressed in GBM and correlated with poor prognosis. The silencing or inactivation of PRMT2 inhibits GBM cell growth and glioblastoma stem cell self-renewal in vitro, and suppresses orthotopic tumor growth, accompanied with significant deregulation of genes mainly associated with cell cycle progression and pathways in cancer. Mechanistically PRMT2 is responsible for H3R8 asymmetric methylation (H3R8me2a), whose enrichment at promoters and enhancers is closely correlated with known active histone marks and is required for the maintenance of target gene expression. Together, this study demonstrates that PRMT2 acts as a transcriptional co-activator for oncogenic gene expression programs in GBM pathogenesis and provides a rationale for PRMT2 targeting in aggressive gliomas.

[1] Department of Cell Biology, Tianjin Medical University, 2011 Collaborative Innovation Center of Tianjin for Medical Epigenetics, Tianjin Key Laboratory of Medical Epigenetics, Qixiangtai Road 22, Tianjin 300070, China. [2] Department of Neurosurgery, Tianjin Medical University General Hospital, Tianjin 300052, China. [3] Laboratory of Neuro-Oncology, Tianjin Neurological Institute, Department of Neurosurgery, Tianjin Medical University General Hospital and Key Laboratory of Neurotrauma, Variation, and Regeneration, Ministry of Education and Tianjin Municipal Government, Tianjin 300052, China. [4] Hubei Key Laboratory of Cell Homeostasis, College of Life Sciences, Wuhan University, Wuhan 430072, China. [5] Department of Neuropathology, Beijing Neurosurgical Institute, Capital Medical University, 6 Tiantanxi Li, Beijing 100050, China. [6] Department of Pathology, Tianjin Nankai Hospital, Tianjin 300100, China. [7] School of Pharmaceutical Sciences, Xiamen University, Xiamen, Fujian 361102, China. These authors contributed equally: Feng Dong, Qian Li, Chao Yang Correspondence and requests for materials should be addressed to C.K. (email: kang97061@gmail.com) or to X.W. (email: wuxudong@tmu.edu.cn)

The low-grade gliomas (World Health Organization (WHO) grade II and III astrocytomas, oligodendrogliomas) are well-differentiated but may proceed to higher grade (grade IV) over time. Glioblastoma multiforme (GBM) is the most common and aggressive form of malignant astrocytoma (grade IV), with a median survival time of 15 months following diagnosis. Improved therapeutic options for high-grade gliomas are urgently needed. Glioblastoma is the first cancer studied by The Cancer Genome Atlas (TCGA; http://www.cbioportal.org) project and a large amount of genomic and transcriptomic data have contributed to the understanding of this lethal disease[1,2]. Similar to that in other cancer types, epigenomic alterations occur in parallel with genetic changes in GBM, leading to deregulated transcriptional programs[3–5]. Given the reversibility of epigenetic changes, identification of key driver chromatin modifiers and a better understanding of the regulatory mechanisms in GBM tumorigenesis will hopefully provide effective therapeutic strategies.

Histones are integral components of chromatin in eukaryotic cells. Diversity of posttranslational modifications on histones and proper combinations are responsible for precise regulation of gene transcription. In the past decades, lysine methylations on histones and the lysine methyltransferases (KMTs) have been intensively studied. In contrast, the roles of arginine methylations on histones are far less known. Three types of methylarginine species exist: ω-NG-monomethylarginine (MMA), ω-NG,NG-asymmetric dimethylarginine (ADMA), and ω-NG,N'G-symmetric dimethylarginine (SDMA)[6,7]. Histone arginine methylations have emerged as one type of important histone modifications involved in transcriptional regulation. For example, H3R2 symmetric dimethylation (H3R2me2s) enhances WDR5 binding and is correlated with H3K4me3 at active promoters[8,9], whereas H3R2 asymmetric dimethylation (H3R2me2a) acts as a repressive mark abrogating the trimethylation of H3K4 (H3K4me3) by the Set1 methyltransferase[9–11]. H4R3me2a[12–14], H3R17me2a[14,15], and H3R42me2a[16] are usually regarded as active marks. It is important to note that these conclusions are either based on the crosstalks with known histone modifications or the expression of individual target genes[6,7]. A clear link of histone arginine methylations to in vivo transcriptional activtiy is far from being established, mainly due to limited knowledge of their genome-wide distribution patterns. Through chromatin immunoprecipitation sequencing (ChIP-seq) analyses, a recent study demonstrated that H4R3me2s is enriched at GC-rich regions independent of transcriptional activity[17], although it has been generally thought as a repressive mark[18–20]. Thus, a complete mechanistic understanding of histone arginine methylations in transcriptional regulation remains to be defined.

Protein arginine methylations are catalyzed by protein arginine methyltransferases (PRMTs), which transfer a methyl group from S-adenosyl-methionine (SAM) to an arginine guanidine nitrogen. These enzymes are divided into three subclasses based on the methylarginine produced. Type I (PRMT1, PRMT2, PRMT3, CARM1/PRMT4, PRMT6, and PRMT8) catalyze to form MMA and ADMA; Type II (PRMT5 and PRMT9) catalyze the formation of MMA and SDMA; Type III catalyzes the formation of MMA only. PRMT7 is either a Type II or Type III enzyme[6]. The deregulated expression of PRMT gene family members has been linked to a variety of diseases including cancers[7]. However, systematic investigations of PRMT members in malignant gliomas are still lacking. So far, PRMT1[12,21] and PRMT5[22–25] have been shown to be indispensable for GBM development. However, little is known about the transcription regulatory mechanisms and molecular networks that underpin the oncogenic functions mediated by these individual PRMTs.

In the present study, we identify PRMT2 as a pro-tumorigenic factor in malignant gliomas and provide evidence that GBM cell growth and tumorigenesis is sensitive to PRMT2 depletion or inactivation in vitro and in vivo. Molecularly, PRMT2-mediated H3R8me2a is responsible for the activation of oncogenic transcriptional programs. Moreover, our findings provide mechanistic insights into an H3R8me2a integrated epigenetic regulatory network in GBM and suggest therapeutic strategies for this highly aggressive subtype of malignancy.

## Results

**PRMT2 expression is elevated in glioblastoma.** We sought to identify PRMT family members that are overexpressed in high-grade gliomas and are potential drivers of GBM pathogenesis. To get insight into the expression profilings of catalytically active PRMTs in glioma samples and their values in prognosis, we first took advantage of TCGA datasets of gliomas (WHO grade II–IV). We found that the expression levels of PRMT1, PRMT2, PRMT4, and PRMT6 are positively correlated with the tumor grades. The expression levels of the other members are either of no significant correlation or negatively correlated with glioma grade (Fig. 1a). In addition, the elevated expression of PRMT1, PRMT2, PRMT4, and PRMT6 predicts poor prognosis in glioma patients. In contrast, the high expression levels of PRMT5, PRMT7, PRMT8, and PRMT9 are correlated with favorable prognosis (Fig. 1b). To avoid bias caused by race differences, we also queried Chinese Glioma Genome Atlas (CGGA) datasets. Interestingly, we observed almost exactly the same patterns as in TCGA glioma datasets (Supplementary Fig. 1a,b). Interestingly, the expression levels of PRMT1, PRMT2, PRMT4, and PRMT6 in low-grade gliomas are significantly higher in *IDH1/2* wild-type (WT) subgroups than the subgroups with the *IDH1/2* mutations. Hence, their high expression may contribute to the malignant progression of gliomas with WT-*IDH* genes (Supplementary Fig. 1c), which usually predicts worse prognosis[26].

To understand which of the PRMT family members are required for the proliferation of GBM cells, we efficiently knocked down each of them respectively in two cell lines with different genetic backgrounds, T98G (*PTEN* WT) and U87 (*PTEN* deleted) (Supplementary Fig. 2). The MTS (3-(4,5-dimethylthiazol-2-yl)-5-(3-carboxymethoxyphenyl)-2-(4-sulfophenyl)-2H-tetrazolium) cell proliferation assays showed that the cell growth is consistently inhibited by the downregulation of PRMT2 in both cell lines. The effects of knocking down other PRMT members vary depending on the GBM cell line (Fig. 1c). Thus, we focus our following investigations on PRMT2 in GBM pathogenesis.

Through the survival analysis in GBM (WHO grade IV), we found that the high expression levels of PRMT2 are significantly correlated with unfavorable prognosis of patients in all datasets that we analyzed (TCGA, CGGA, REMBRANDT) (Fig. 1d). To avoid the bias from the analysis of only mRNA expression data, we examined the PRMT2 protein levels in vivo by immunohistochemistry (IHC) staining of 21 cases of resected tumor samples representing different grades of glioma (the clinical information of the samples is listed in Supplementary Table 1). As shown in Fig. 1e, we observed a clear nuclear enrichment of PRMT2 in high-grade gliomas. Quantification of the staining (see Methods) revealed a strong association between the abundance of PRMT2-positive cells and higher tumor grade ($p < 0.001$; Fig. 1f). Therefore, the elevated expression level of PRMT2 is an indicator of the aggressiveness of malignant gliomas.

**PRMT2 is required for the GBM cell growth.** To more carefully elucidate the role of PRMT2 in the survival and proliferation of GBM cells, we knocked it down with two different specific short

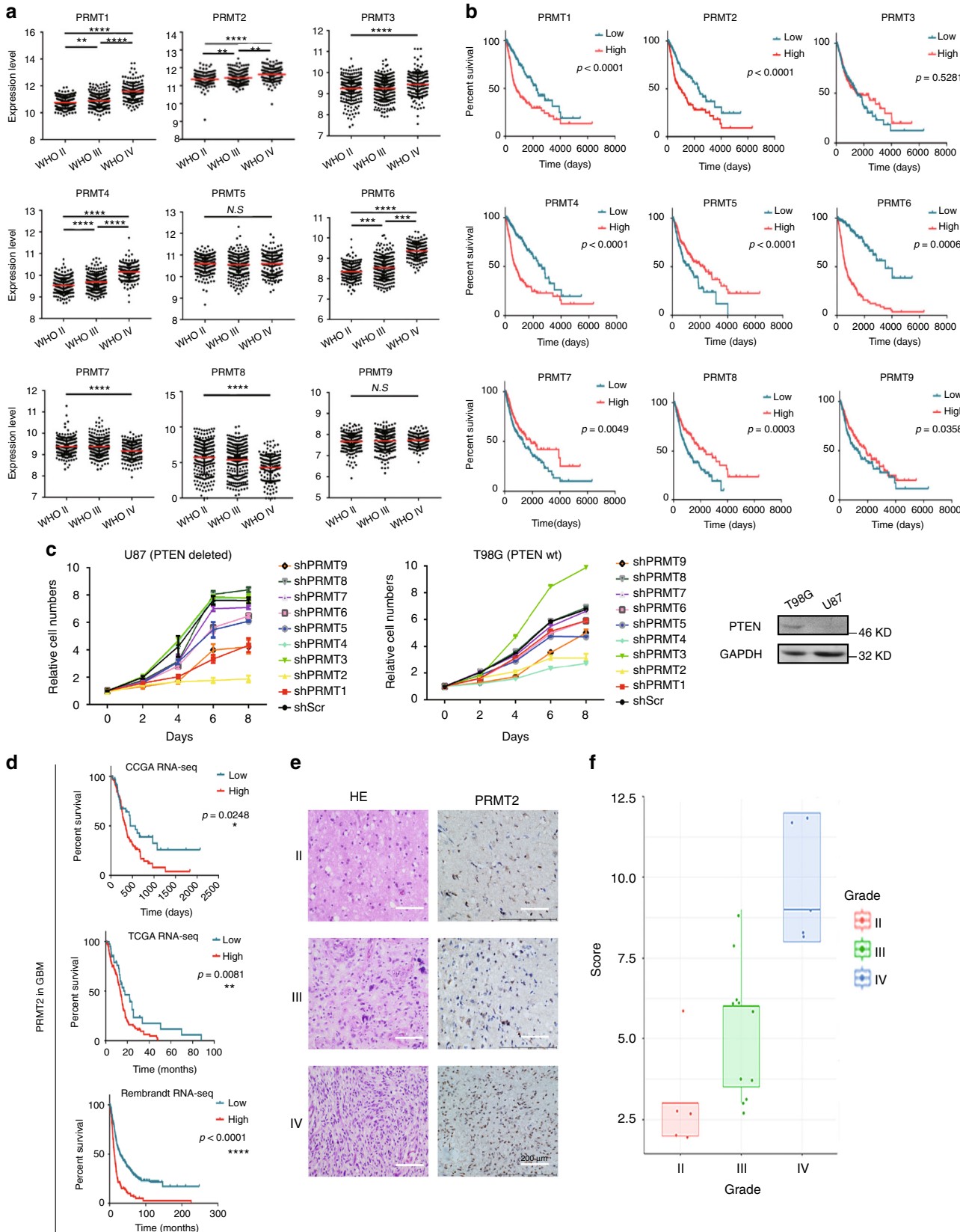

hairpin RNAs (shRNAs) (shPRMT2-1 and shPRMT2-2) in U87 and T98G cells. Upon an effective depletion of PRMT2 as tested by real-time quantitative PCR (RT-qPCR) and western blot (WB) assays (Fig. 2a, b), the cell growth in either colony formation in soft agar or monolayer culture, is significantly inhibited (Fig. 2c, d). Notably,

PRMT2 depletion does not trigger cellular apoptosis as detected by flow cytometry analysis of Annexin-V-stained U87 cells (Supplementary Fig. 3a and b, and see Supplementary Methods). To avoid the bias of immortalized GBM cell lines cultured in serum medium, we also used two tumor progenitor cell (TPC) lines (TPC1115 and

**Fig. 1** PRMT2 expression is elevated in glioblastoma and associated with adverse prognosis. **a** The mRNA levels of PRMT family members were analyzed in different grades of gliomas according to the TCGA datasets. Significance level was determined using one-way ANOVA followed by Dunnett's multiple comparisons test. *$p \leq 0.05$, **$p \leq 0.01$, ***$p \leq 0.001$, and ****$p \leq 0.001$. **b** Kaplan–Meier survival curves for correlation between mRNA expression of each PRMT family member and survival of glioma patients in the TCGA dataset. **c** Cell growth curves of two GBM cells expressing the indicated shRNAs. MTS assay was employed at the specified time points to follow the rate of proliferation. The error bars denote SD, $n = 3$. WB analysis validates the differential PTEN expression status. GAPDH serves as a loading control. **d** Kaplan–Meier survival curves for correlation between PRMT2 mRNA expression and survival of GBM patients in the TCGA RNA-seq dataset, CGGA RNA-seq dataset, or Rembrandt RNA-seq dataset. **e** Representative examples from IHC analysis of PRMT2 protein levels in different grade of glioma specimens are shown. Scale bar, 200 μm. **f** Correlation between PRMT2 protein levels and malignant degree of gliomas. Tumor sections from 21 glioma specimens were IHC-stained with anti-PRMT2 antibody. Lines within boxes indicate medians of the scores

0411) from surgically resected glioblastoma samples[27]. The two lines are respectively phosphatase and tensin homolog (PTEN) deleted and WT (Supplementary Fig. 4a). Growth curves demonstrated that the depletion of PRMT2 (Supplementary Fig. 4b) also significantly attenuates their growth in serum-free monolayer culture condition (Fig. 2e). These finding combined show that the anchorage-dependent and -independent growth of GBM cells is sensitive to PRMT2 depletion in vitro.

**PRMT2 depletion reduces GSC self-renewal and tumorigenesis.** It is well recognized that a minor population of tumor cells, known as cancer stem cells (CSCs) or tumor-initiating cells (TICs), are responsible for the high aggressiveness and tumor relapse after chemo- and/or radiotherapy. Eradicating these cells has been shown to improve the therapy efficacy[28,29]. Therefore, we assessed how PRMT2 depletion affected the self-renewal of glioblastoma stem cells (GSCs). As GSCs tend to form spheres when grown in suspension in defined serum-free media, we first performed a limiting dilution assay for sphere formation with the two cell lines transduced with shPRMT or shScr. As shown in Supplementary Fig. 5a and Fig. 3a, b, the PRMT2-depleted U87 and T98G cells give rise to fewer and smaller spheres than the similarly cultured control cells. Considering patient-derived cells better reflect key properties of GSCs such as differentiation capabilities and heterogeneity, we then used TPC1115 and 0411 to measure the sphere-formation frequency. Clearly, the TPC lines have much higher capacity to form spheres than the two cell lines. In addition, the sphere re-initiating cell frequency in the PRMT2 knockdown group is much lower compared with controls (Supplementary Fig. 5a and Fig. 3c). To further confirm the effects of PRMT2 depletion on GSC self-renewal, we examined the expression of stem cell-associated genes by quantitative reverse-transcriptase PCR (qRT-PCR) analysis. As shown in Supplementary Fig. 5b, the representative stem cell genes (e.g., CD133, OCT4, and SOX2) are markedly downregulated by PTMT2 knockdown in the two TPCs. These data indicate that PRMT2 is required for the maintenance of GSCs in vitro.

Next, we proceeded to look into the impact of PRMT2 downregulation on GBM growth in vivo. We established orthotopic models using U87 and TPC1115 cells transduced with luciferase expressing lentivirus. Then U87-luciferase cells ($2 \times 10^5$) and TPC1115-luciferase cells ($2 \times 10^4$) were injected intracerebrally. For both models, PRMT2 downregulation results in a significant reduction of tumor volumes as detected by the bioluminescence imaging (Fig. 3d) and quantification of the luciferase activity in implanted mice brains at different time points (Fig. 3e). Intracranial space-occupying lesions were obviously detected by the magnetic resonance imaging scanning in the control group. In contrast, the PRMT2-depleted tumors only mildly affect normal brain structure (Supplementary Fig. 5c, and see Supplementary Methods). Compared with nude mice injected with control U87 cells, the mice bearing tumors derived from PRMT2-depleted U87 cells show significantly prolonged

survival by Kaplan–Meier survival curves (Fig. 3f). Morphologically, hematoxylin–eosin (HE)-stained slides indicated apparent invasive phenotype in control tumors as illustrated by infiltrating tumor cells into the neighboring normal brain tissue and palisading around necrotic foci in the margin. In contrast, the PRMT2 knockdown leads to barely visible tumors in the brain sections or inhibits the tumor invasiveness (Fig. 3g). These data suggest that PRMT2 is indispensable for GBM pathogenesis, and that it may serve as a potential therapeutic target for GBM.

**PRMT2 depletion affects oncogenic transcriptional programs.** To decipher how PRMT2 downregulation alters the transcriptional program in GBM cells, we undertook an unbiased genomic approach to define the transcriptional program controlled by PRMT2. We performed RNA sequencing (RNA-seq) to identify transcripts differentially expressed upon PRMT2 knockdown in U87 (shScr vs. two independent shPRMT2 sequences). Bioinformatics analyses revealed that the deregulated genes by the two shRNAs largely overlap and the number of downregulated genes are around three times of the upregulated gene numbers in each of shPRMT2 group (fold change > 2) (Fig. 4a, b, Supplementary Data 1). Kyoto Encyclopedia of Genes and Genomes (KEGG) analysis demonstrated that the overlapped 2407 downregulated genes are significantly associated with cell cycle, DNA replication, pathways in cancer, etc (Fig. 4c). Further analysis of TCGA RNA-seq dataset for GBM identified 583 genes whose expression levels are positively correlated with PRMT2 expression ($p < 0.001$ and correlation coefficient > 0.3). Using these PRMT2-positive correlated genes as the defined gene set, gene-set enrichment analysis (GSEA) showed that PRMT2 positively correlated genes are significantly enriched in the control group (Scr), compared with PRMT2-depleted group (KD) (false discovery rate (FDR) $q$-value = 0.045) (Fig. 4d). The expression of several representative genes associated with cell cycle (CCNA2, CCNB1/2, CCND1, CDK4/6) and signaling pathways (AXIN2, DKK3) were validated in U87 cells by qRT-PCR analysis (Fig. 4e). Consistently, IHC staing of CCNB1, CCND1, and CDK4 of the 21 glioma specimens confirmed that the protein expression levels of these cell cycle gene are correlated with PRMT2 in different grades of gliomas (Supplementary Fig. 6a and b). The changes at the protein levels were further confirmed by WB assays. As illustrated in Fig. 4f, various oncogenic signaling pathways such as PI3K-AKT, MAPK, JAK-STAT, and Wnt/β-catenin pathways are strikingly inhibited in PRMT2-depleted U87 cells. These effects are partly supported by previous findings showing that PRMT2 functions as a co-activator for β-catenin[30,31].

Similarly, we also performed RNA-seq analysis in PRMT2-depleted vs. control TPC1115 cells. In contrast to 698 commonly upregulated genes by two shRNAs, three times of downregulated genes (2088) were identified (Supplementary Fig. 7a, Supplementary Data 2). Furthermore, 431 downregulated genes are shared in TPC1115 and U87 cells (Supplementary Fig. 7b). Among the oncogenic signaling pathways, the JAK-STAT signaling pathway,

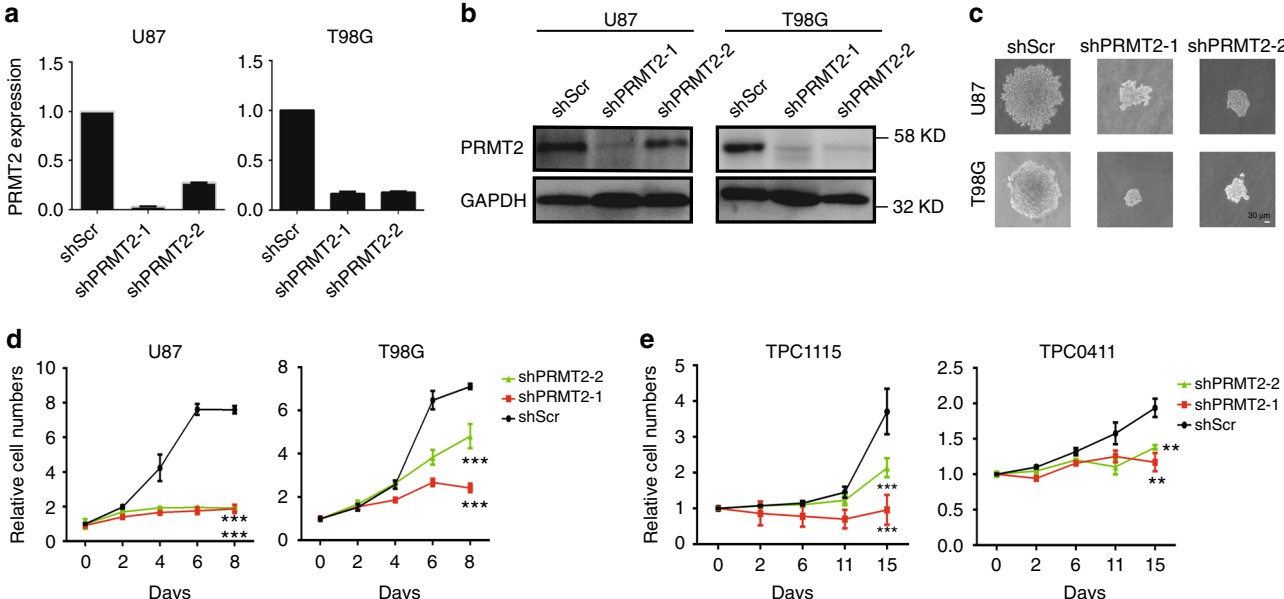

**Fig. 2** PRMT2 is required for GBM cell anchorage-dependent and -independent growth. **a**, **b** PRMT2 expression levels in U87 and T98G cells transduced with two different shRNA constructs targeting PRMT2 (shPRMT2-1 and shPRMT2-2) or scramble shRNA (shScr) were determined by RT-qPCR (**a**) and WB analysis (**b**). GAPDH serves as a loading control. The error bars in **a** denote SD, $n = 2$. **c** Anchorage-independent growth of U87 and T98G cells transduced with shScr or shPRMT2 was measured by colony formation in soft agar. Representative colonies were photographed after 4 weeks. Scale bar, 30 μm. **d**, **e** Cell growth curves of U87 and T98G cells (**d**), TPC1115 and TPC0411 cells (**e**) expressing the indicated shRNAs. MTS assay was employed at the specified time points to follow the rate of proliferation. Significance level was determined using Student's two-sided t-tests. $**p \leq 0.01$ and $***p \leq 0.001$. Error bars, ± SD, $n = 3$

which is critical for brain tumor development, is strikingly inhibited in PRMT2-depleted TPC1115 cells as shown by the GSEA (Supplementary Fig. 7c) and WB analyses (Supplementary Fig. 7d). These data indicate that PRMT2 is required to sustain the oncogenic gene expression programs crucial for GBM maintenance.

**PRMT2 is responsible for the maintenance of H3R8me2a levels**. To test whether the altered gene expression could be a consequence of loss of PRMT2-mediated histone arginine methylation, we first compared the global histone methylation levels in control and PRMT2 knockdown U87 cells. We found that the levels of asymmetric dimethylation on H3R8 (H3R8me2a) are specifically decreased in the PRMT2-depleted cells as shown by the WB and immunofluorescence (IF) assays (Fig. 5a, b). This is consistent with previously reported in vitro H3R8me2a methylation activity using either peptides or recombinant histones as substrates[30,31]. Although PRMT2 alone did not show strong activity in vitro, it is indispensable for the maintenance of H3R8me2a in vivo.

We next explored the distribution of H3R8me2a in the genome through an optimized ChIP protocol (see Methods). In total, the ChIP-seq analysis in U87 cells identified 13,078 H3R8me2a-enriched regions. Most of these are distributed at the intragenic or intergenic regions and 13.71% at promoters, within a window of ± 3 kb from the transcription start site (TSS) (Fig. 5c). Moreover, we found that the enrichment levels of H3R8me2a are significantly decreased in PRMT2-depleted cells compared to control (Fig. 5d), further confirming the specificity of the antibody. These data support the responsibility of PRMT2 for the maintenance of H3R8me2a levels on chromatin.

**H3R8me2a is important for the activation of target genes**. To further make clear of the roles of PRMT2 in transcriptional regulation, we tried to identify PRMT2 target genes by ChIP-seq. Unfortunately, we could not get access to a ChIP-grade antibody.

As PRMT2 is the main resource for H3R8me2a in vivo (Fig. 5a), we define the regions with decreased H3R8me2a signals in PRMT2-depleted cells as PRMT2 target sites. Hence, we turned to examine how the loss of PRMT2-mediated H3R8me2a correlates with changes of the target gene expression as detected by the RNA-seq analysis. Focusing on the PRMT2-dependent H3R8me2a peaks ( > 30% decrease by PRMT2 depletion) distributed at the promoters (699 peaks), distal intergenic regions (3078 peaks), and gene bodies (1422 peaks), we compared the expression of host or nearby genes in control and PRMT2-depleted cells. Interestingly, accompanied with the decreased H3R8me2a levels at either promoters or inter/intragenic regions, the associated genes are significantly downregulated in PRMT2-depleted cells (Fig. 5e). This may also explain why PRMT2 depletion mainly results in gene downregulation (Fig. 4a and Supplementary Fig. 6a). For example, peaks identified by the peak-calling software MACS confirm the decreased H3R8me2a levels on chromatin are correlated with the downregulated expression of genes such as AXIN2, CDK6 and TIAM1 (Fig. 5f). These data indicate that PRMT2-dependent H3R8me2a deposition is required for transcriptional activation of its target genes.

**Decreased H3R8me2a affects active promoters and enhancers**. H3R8me2a has been shown to antagonized G9 methylation activity on H3K9 in vitro[32]; however, little is known about its in vivo functions in transcriptional regulation. To capture a clearer picture of this not well-characterized histone mark, we compared the H3R8me2a enrichment profile with the genomic distributions of well-known active histone modifications (H3K4me1, H3K4me3, and H3K27ac) and repressive histone modifications (H3K27me3 and H3K9me3). As shown in the heatmaps (Fig. 6a), H3R8me2a-enriched promoters (1793 peaks) are decorated with H3K4me3 but negative with H3K27me3, characteristic of active promoters. At distal intergenic regions or gene bodies, H3R8me2a peaks (5247 and 5947 peaks) are closely

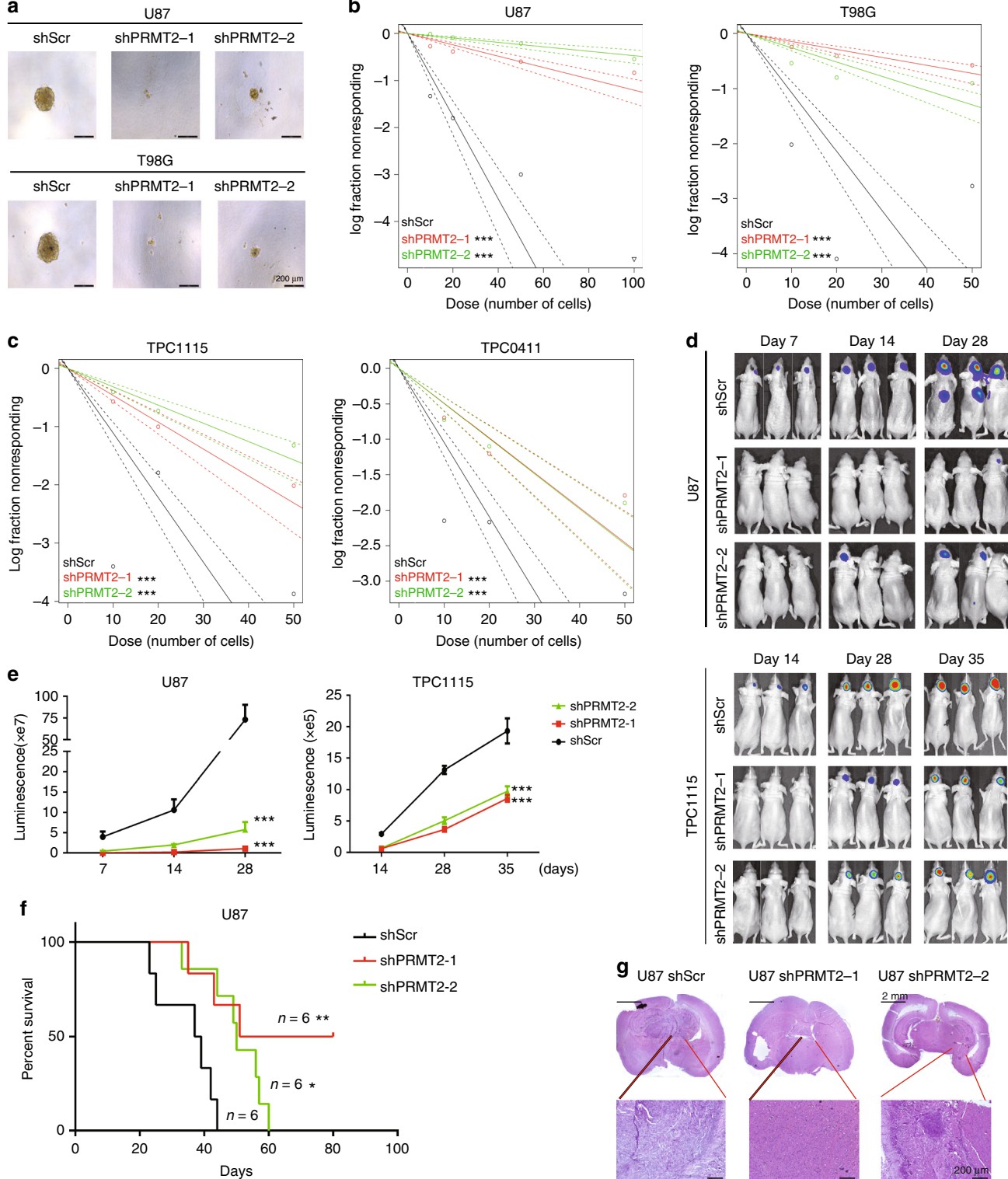

overlapped with H3K4me1 and H3K27ac-rich sites, characteristic of active enhancers. In contrast, none of these target sites are correlated with H3K27me3 or H3K9me3 (Fig. 6a). These correlation analyses further demonstrate that H3R8me2a is closely associated with active histone marks while anti-correlated with repressive histone marks.

Next we focused on understanding the role of PRMT2 on H3K4me1, H3K4me3, and H3K27ac levels at PRMT2-dependent H3R8me2a peaks (6698 peaks). As shown in Fig. 6b, all these active histone modifications levels are significantly decreased

upon depletion of PRMT2, especially for H3K4me3 and H3K27ac. Examples of ChIP-seq enrichment profiles for the indicated loci are shown in Fig. 6c and Supplementary Fig. 8. These effects were further validated by the ChIP-qPCR analysis at the promoters or enhancers of the downregulated genes as previously confirmed by RT-qPCR analysis (Fig. 4e). Notably, the decrease of active histone modification levels accompanied with the loss of H3R8me2a upon depletion of PRMT2 are specific, because no significant changes were observed for the house-keeping gene *GAPDH* (Fig. 6d). To further confirm these affected

**Fig. 3** PRMT2 is required for the maintenance of GSCs and tumorigenesis. **a** Representative images of tumor spheres in dose of 10 cells/well are shown. Scale bar, 200 μm. **b**, **c** In vitro limiting dilution assays plating decreasing number of GSCs with or without PRMT2 knockdown calculated with extreme limiting dilution assay analysis in U87 and T98G cells (**b**), and TPC1115 and TPC0411 cells (**c**) (mean ± SD, $n = 3$). Frequency and probability estimates were computed using the ELDA software. ***$p \leq 0.001$. **d** Representative luciferase images of three mice per group at 7, 14, and 28 days post tumor implantation. U87 or TPC1115-luciferase cells were transduced with shScr or shPRMT2. Color scale for U87 cells: Min $= 5.00 \times e^6$, Max $= 5.00 \times e^8$ (top panel); color scale for TPC1115 cells: Min $= 5.00 \times e^4$, Max $= 2.00 \times e^5$ (bottom panel). **e** The average signals of luciferase activity in implanted mice brains at different time points were respectively compared in U87 and TPC1115 cells transduced with shScr or shPRMT2 (mean ± SD; $n = 6$). Significance level was determined using Student's two-sided $t$-tests. ***$p \leq 0.001$. **f** Survival analysis of mice intracranially implanted with U87 cells with or without PRMT2 knockdown. X axis represents days after cells injection. Significance level was determined using log-rank analysis. *$p \leq 0.05$, **$p \leq 0.01$. $n = 6$ for each treatment group. **g** Representative images of H&E staining for tumor formation in control (shScr) and PRMT2-depletion group (shPRMT2). Top panel: scale bar, 2 mm. Bottom panel: scale bar, 200 μm

regions are indeed PRMT2 target sites, we generated a tetracycline-inducible Flag-tagged PRMT2 expression system in U87 cells and did Flag ChIP-qPCR analysis. As shown in Supplementary Fig. 9a, the addition of Doxycycline (Dox) clearly induces the ectopic expression of Flag-PRMT2 enrichment. Accordingly, the enrichment of Flag-PRMT2 is significantly induced by Dox at the target enhancers and promoters, but not at the *GAPDH* promoter (Supplementary Fig. 9b). Taken together, PRMT2-mediated H3R8me2a is required for the maintenance of active promoters and enhancers of the PRMT2 target genes.

**PRMT2's pro-tumorigenic functions is activity dependent**. To test how H3R8me2a levels correlate with the tumor pathogenesis, we performed IHC staining of the clinical glioma samples. As shown in Fig. 7a, H3R8me2a levels are significantly higher in high-grade gliomas. Moreover, the paired sample analysis showed that the signals of H3R8me2a are positively correlated with the intensity of Ki67 staining, a cellular marker for proliferation (Fig. 7b). This is also consistent with the transcriptional outcome of PRMT2-mediated H3R8me2a on cell cycle progression.

To examine whether the catalytic activity of PRMT2 is required for GBM tumorigenesis, we generated two PRMT2 catalytic inactive mutants H112Q and M115I, based on the prediction (see Supplementary Methods) of the pocket site for SAM docking on PRMT2[33] (Supplementary Fig. 10a) and cloned them into tetracycline-inducible expression vectors. Upon successful transduction of each of them into U87 and T98G cells, only the induced expression of PRMT2-H112Q was found to specifically affect H3R8me2a levels in both cell lines (Fig. 7c and Supplementary Fig. 10b). In vitro cell proliferation and sphere-formation assays confirmed that the cell growth and sphere-formation efficiency were significantly inhibited by the induced PRMT2-H112Q expression, but not by the PRMT2-M115I expression (Fig. 7d, e, Supplementary Fig. 10c). Furthermore, the induced expression of inactive PRMT2 (H112Q) specifically disrupted the expression of cell cycle regulated genes (Fig. 7f and Supplementary Fig. 10d) and reduced the in vivo tumor growth (Fig. 7g and Supplementary Fig. 10e). The Kaplan–Meier curves indicate significant longer survival of mice bearing PRMT2-H112Q expressing tumors compared with the control (Fig. 7h). Dissecting the xenografted tumors from each group for the IHC staining, we found that induced PRMT2-H112Q-expressing tumors displayed reduced H3R8me2a activity and lower Ki67 density compared with the uninduced control group (Fig. 7i). In conclusion, these results support that PRMT2 contributes to GBM tumor growth through its catalytic activity.

## Discussion
It is well recognized that epigenetic dysregulation allows rapid selection for tumor cell survival. As a consequence,

transcriptional dependencies develop and provide opportunities for novel therapeutic interventions in cancer. In this study, we show that PRMT2 is a pro-tumorigenic factor in GBM. Moreover, we demonstrate that PRMT2-mediated H3R8me2a is a critical modification for active promoters and enhancers, which underlies at least part of the oncogenic transcriptional programs in GBM. Given the possibility of its prevalence in other malignancies, our findings may have even broader implications for the control of cancers.

Although genetic alterations of the *PRMT* gene family are rarely observed, upregulation of these enzymes have been frequently found in cancers. A large amount of studies has been focusing on the roles and mechanisms of PRMT5 in malignancies including malignant gliomas[7,22–25,34]. Combined with clinical data analyses and loss-of-function assays, here we confirm that PRMT2 is indispensable for the development of malignant glioma. The GBM cell proliferation, invasiveness, and tumorigenesis is sensitive to PRMT2 silencing or inactivation (Figs. 2, 3, and 7).

Unlike KMTs, PRMT enzymatic activities remain somewhat enigmatic. Previous results have shown that PRMT2 can form a heterodimer with PRMT1, thereby stimulating its activity on H4R3me2a[35]. However, we did not find detectable effects on H4R3me2a after the downregulation of PRMT2 (Fig. 5a). Instead we show that PRMT2 is specifically responsible for H3R8me2a, which is not affected by PRMT1 depletion (data unpublished). Thus, our data suggest that PRMT2 could exert its methyltransferase activity independent of PRMT1. Besides, there remains gaps between PRMT activities and its biological functions. Despite of growing ChIP-seq datasets for a diversity of histone marks, qualified data for histone arginine methylation marks lack so far, probably either due to low efficiency or specificity of antibodies available, or due to unsuitable methods used. To understand the roles of PRMT2-catalyzed H3R8me2a, we optimized a ChIP method for the specific H3R8me2a antibody and managed to obtain a qualified dataset. For the first time, we reveal the genomic localization of H3R8me2a and link this critical modification to active chromatin. Furthermore, we provide direct evidence that H3R8me2a is essential for the in vivo maintenance of chromatin modifications at active promoters and enhancers. Nevertheless, we still do not understand how H3R8me2a is recognized and thereby can affect other epigenetic modifications. Interestingly, a dual mark H3K4me3R8me2a is recognized by Spindlin1 in vitro[31]; however, the existence in vivo and its biological significance remains unknown. It is worth following to find out how H3R8me2a crosstalks with other modifications and stimulates the downstream oncogenic gene expession for tumor progression. In addition, it is noteworthy that PRMT2 silencing does not result in cell apoptosis, although it attenuates cell growth. Hence, further mechanistic understanding will hopefully facilitate the development of optimal combinatorial therapeutic strategies.

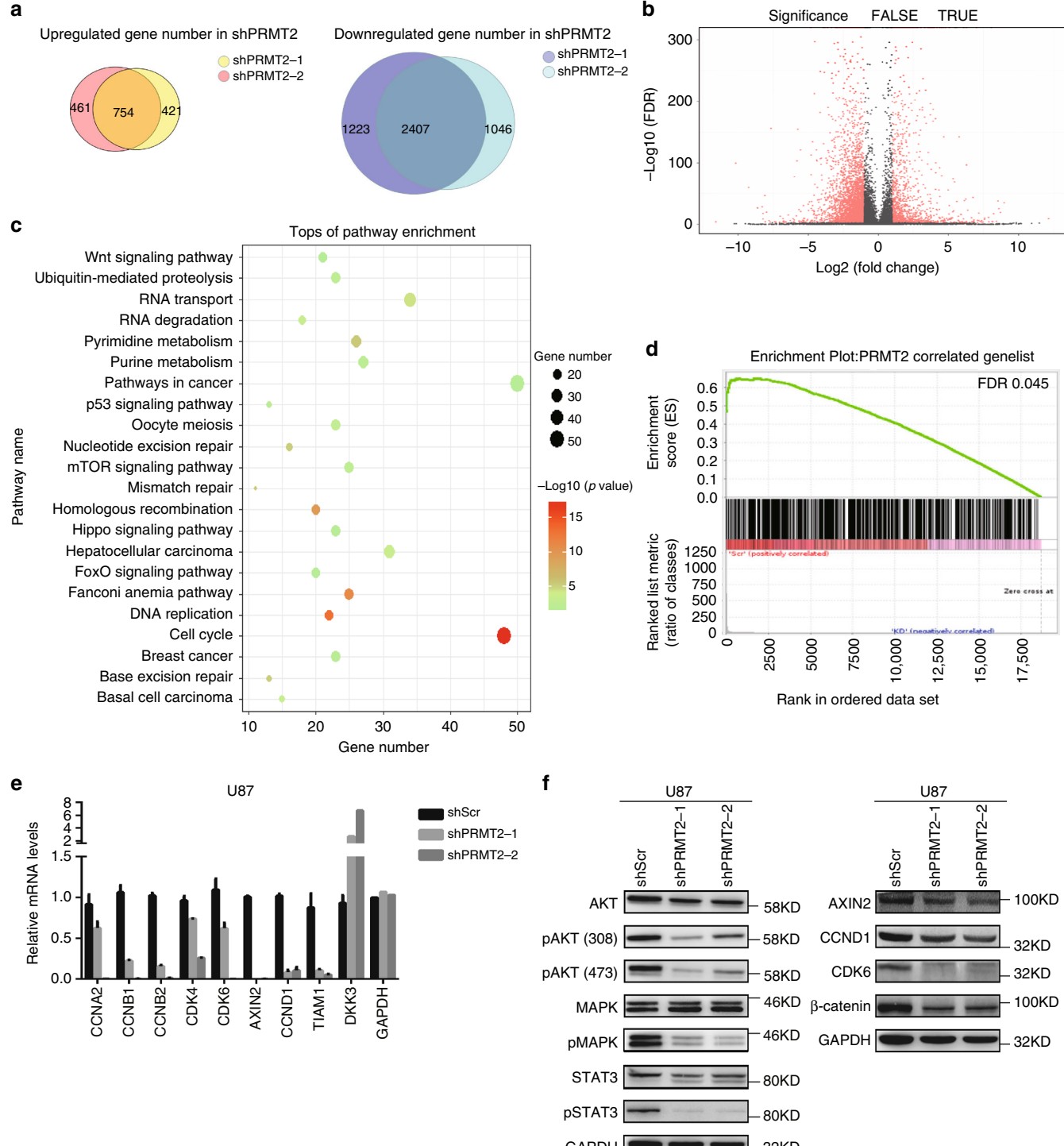

**Fig. 4** PRMT2 is essential for the activation of oncogenic transcriptional programs. **a** Venn diagrams show 754 upreglated genes with overlap in U87 shPRMT2-1 compared with shPRMT2-2 (top panel) and 2407 downregulated genes with overlap in U87 shPRMT2-1 compared with shPRMT2-2 (bottom panel). **b** Volcano plot shows the transcript levels differentially expressed between U87 shScr and U87 shPRMT2. Filtered by $\log_2$ (fold change) $\geq 1$ and false discovery rate (FDR) $< 0.001$, significantly dysregulated genes are shown as red dots. The representative genes involved in cell cycle progression and pathways in cancer are labeled. Upregulated genes number = 1215, whereas downregulated genes number = 3630. **c** KEGG pathway enrichment analyses of significantly downregulated genes in U87 shPRMT2. Tops of enriched pathway are shown. **d** GSEA indicates PRMT2 positively correlated genes (583 genes, $p < 0.001$ and $r > 0.3$)) in the TCGA dataset is significantly enriched in the control group (Scr), compared with PRMT2-depleted group (KD). **e** RT-qPCR analysis of deregulated genes involved in cell cycle progression and pathways in cancer in U87 cells transduced with shScr or shPRMT2. Error bars ± SD, $n = 2$. **f** WB analyses evaluating cell cycle genes and oncogenic pathways in U87 cells expressing shScr or shPRMT2. GAPDH was used as loading control

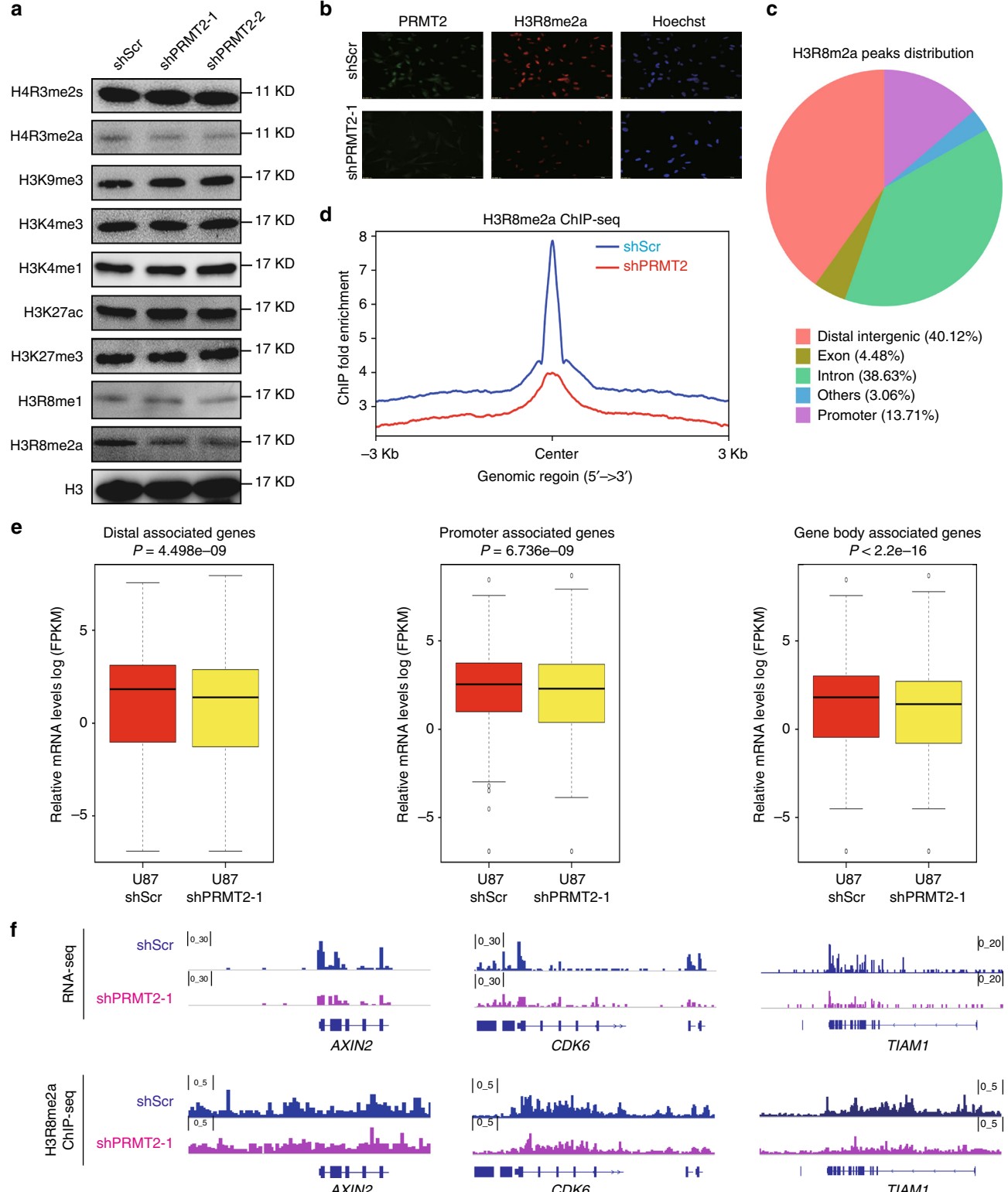

**Fig. 5** PRMT2 is responsible for the maintenance of H3R8me2a levels and target gene activation. **a** WB analysis of designated histone modification levels in U87 cells expressing shScr or shPRMT2. Histone 3 was used as loading control. **b** Representative IF images of U87 cells with PRMT2 and H3R8me2a antibody. Scale bar, 50 μm. **c** Piechart shows the proportion of total H3R8me2a peaks in the indicated regions. **d** Average profiles of H3R8me2a across a genomic window of ± 3000 bp surrounding the peak summit in shScr an shPRMT2. H3R8me2a peak number 13,078 in U87 shScr, 11,374 in U87 shPRMT2. **e** Comparison of PRMT2 target gene expression in U87 cells expressing shScr and shPRMT2. The PRMT2 target genes are divided into three subgroups (promoter, distal, gene body) according to the distribution of PRMT2-dependent H3R8me2a peaks. The bottom and the top of the boxes represent the first and third quartiles respectively. Lines within boxes indicate medians. Whiskers represent the most extreme data within 1.5 times of the interquartile range. Circles represent outliers. The P-values are calculated by Wilcoxon's signed-rank test. **f** Genomic snapshots of RNA-seq analyses (top panel) and H3R8me2a ChIP-seq analyses (bottom panel) of representative genes in U87 shScr and shPRMT2 cells

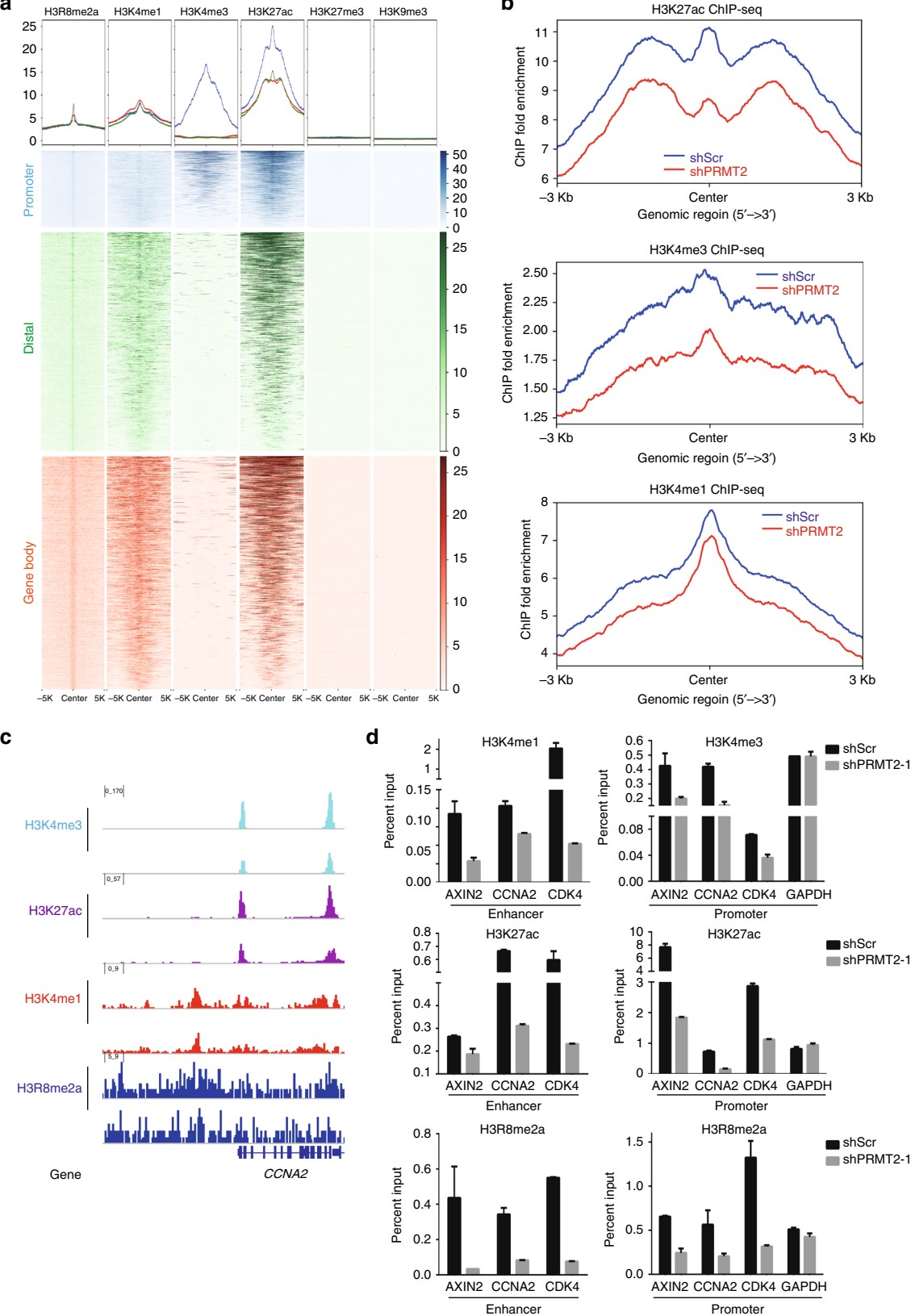

A multitude of chemical inhibitors targeting epigenetic modifiers has been successfully developed in the past decades and many of these are either in clinical or preclinical trials for cancer treatment[36–38]. Several selective inhibitors of PRMT family members have been tested and showed promising effects in a diversity of tumor models[39–43]. These inhibitors usually bind to the sites occupied by the substrate arginine and flanking side chains, and behave as substrate competitors[38]. Notably most of the histone-modifying enzymes (including PRMTs) have multiple non-histone substrates that may also be critical for oncogenesis.

**Fig. 6** PRMT2-mediated H3R8me2a is important for the maintenance of active promoters and enhancers. **a** Average profiles of indicated histone modifications across a genomic window of ±5000 bp surrounding the H3R8me2a peak summit (top panel). Heatmaps of indicated histone modifications across regions of ±5000 bp surrounding the H3R8me2a peak summit in each cluster. According to H3R8me2a peak distribution, H3R8me2a peaks are divided into three clusters (promoter, distal, gene body) (bottom panel). H3R8me2a peak number in promoter region = 1793, in distal region = 5247, in gene body region = 5947. **b** Average profiles of H3K27ac, H3K4me3, and H3K4me1 across a genomic window of ±3000 bp surrounding the PRMT2-dependent H3R8me2a peak (6698 peaks) summit. **c** Genomic snapshots of ChIP-seq analyses for H3K4me3, H3K27ac, H3K4me1, and HER8me2a in U87 cells transduced with shScr (upper track) and shPRMT2 (lower track). **d** ChIP-qPCR analysis of H3K4me3, H3K4me1, H3K27ac, and H3R8me2a enrichment on the promoter or enhancer regions of PRMT2 target genes. *GAPDH* promoter serves as a negative control (error bars, mean ± SD; $n = 3$)

As for PRMT2, it has been implicated in the methylation of E1B-AP5 (hnRNPU-like 1) and STAT3[44,45]. Also considering that PRMT2 may directly participate in multiple cellular pathways in cancers, it will be interesting to further identify potential non-histone substrates in relevant models. Thus, to pursue specific PRMT2 inhibitors as a targeted approach in the future, a careful investigation of the structural basis for its substrate specificity is required.

In summary, our study implicates PRMT2 as a potential bio-marker for predicting the overall survival and as a therapeutic target in GBM patients. Moreover, our data provide molecular insights into the mechanisms how PRMT2 link H3R8me2a, a poorly known histone mark, to the activation of oncogenic gene expression programs. These findings pave the way to revolutionize our therapeutic options by blocking either PRMT2 activity or the associated transcription regulatory network to tame malignant gliomas and even probably other cancers.

## Methods

**Study approval**. All research performed was approved by the Institutional Review Board at Tianjin General Hospital and was in accordance with the principles expressed at the Declaration at Helsinki. Written informed consent was received from all participants. All animal experiments were performed according to Health guidelines of Tianjin Medical University Institutional Animal Use and Care Committee.

**Cloning and plasmid preparation**. Human PRMT2 cDNA was amplified and introduced into Gateway Entry vector pCR8/GW/TOPO (Invitrogen) following the manufacturer's protocol and verified by sequencing. To generate the PRMT2 mutants, we executed site-directed mutagenesis using the QuikChange MultiSite-Directed Mutagenesis Kit (Stratagene). Briefly, DpnI enzyme was added to digest the PCR product for 1 h after mutagenesis reaction, followed by transformation. The successful mutagenesis was verified by sequencing. Different constructs were subcloned in the desired vectors by Gateway technology (Invitrogen). Specific oligonucleotides against human PRMT members were cloned into pLKO.1 TRC cloning vector according to the protocol recommended by Addgene. The sequences for primers or oligos are listed in Supplementary Table 2.

**Cell culture**. The glioblastoma cell lines U87 and T98G were obtained from the American Type Culture Collection (Manassas, Virginia, USA) and cultured in Dulbecco's modified Eagle's medium (DMEM) containing 10% fetal bovine serum (FBS). TPC1115 and TPC0411 were obtained from fresh surgical specimens of human primary GBMs and cultured as either monolayer or tumor spheres[27] in DMEM/F12 medium supplemented with N2, B27 (Gibco), 20 ng/ml human fibroblast growth factor-basic (bFGF, Sino Biological, Beijing, China), 20 ng/ml epidermal growth factor-basic (EGF, Sino Biological, Beijing, China). All cells were maintained at 37 °C in a humid incubator with 5% CO₂. For sphere formation, cells were cultured in serum-free DMEM/F12 medium containing 20 μl/ml B27 supplements, 5 μg/ml insulin, 20 ng/ml bFGF, and 10 ng/ml EGF at ultra-low attachment plates (Corning). Cells have been authenticated by examining their karyotypes and morphologies. All cells have been tested for mycoplasma contamination by PCR and were verified to be mycoplasma free.

**MTS proliferation assay**. Cells were placed in 96-well plates at $1 \times 10^3$ cells/well in 100 μl of growth medium. For the measurement, cells at exponential phase were incubated in dark with 100 μl of MTS solution (Medium:MTS:PMS = 100:20:1) at 37 °C for 1.5 h. The absorbance was measured at 490 nm. Assays were performed in replicates of six wells/condition. PMS: phenazine methosulfate.

**Anchorage-independent growth assays**. Cells were plated in triplicate in six-well plates. GBM cells ($5 \times 10^3$) per well were seeded in DMEM supplemented with 10% FBS containing 0.35% low-melting agarose on the top of bottom agar containing 0.6% low-melting agarose in growth medium. Cells were fed every 3 to 4 days with fresh growth medium. Pictures were taken after 14 days of incubation.

**Limiting dilution analysis**. GBM cells were seeded in 96-well plates containing 100 μl completed medium at different densities. As the sphere-forming capabilities vary among the different cells, we plated 100, 50, 20, 10 for U87 and the derived cells, whereas plated 50, 20, and 10 cells for other cells. Furthermore, each well was examined for the formation of tumor spheres after 14 days. GSC frequency was calculated using extreme limiting dilution analysis (http://bioinf.wehi.edu.au/software/elda/).

**Immunofluorescence**. Cultured cells were fixed with 4% formaldehyde for 5 min, permeabilized with 0.1% Triton X-100 in phosphate-buffered saline (PBS) for 10 min and then incubated with 5 mg/ml bovine serum albumin for 60 min at room temperature. Immunostaining was performed using the appropriate primary antibodies overnight at 4 °C: anti-PRMT2 (1:100; Abcam) and anti-H3R8me2a (1:500; Novus Biologicals). After careful washing, the cells were incubated with FITC 490 secondary antibody (ZSBIO, dilution 1:200) for 1 h at room temperature. Nuclei were counterstained with Hoechst.

**HE and IHC staining**. The paraffin-embedded tissue sections were used for examination of HE staining. For the IHC staining, specimen's tissue slides were de-paraffinized, rehydrated, and antigen retrieval. The tissue slides were blocked for at least 1 h at room temperature and then incubated with appropriate primary antibodies overnight at 4 °C. The antibodies included anti-PRMT2 (1:100; Abcam), anti-CCNB1 (1:200; Santa Cruz Biotechnology), anti-CCND1 (1:100; Absin), anti-CDK4 (1:200; Wanleibio), and anti-H3R8me2a (1:500; Novus Biologicals). After careful washing, the slides were incubated with horseradish peroxidase (HRP) conjugates using DAB detection. For the IHC analysis, we quantitatively scored the tissue sections according to the percentage of positive cells and staining intensity. We assigned the following proportion scores: 1 if 0–25% of the tumor cells showed positive staining, 2 if 26–50% of cells were stained, 3 if 51–75% stained, and 4 if 76–100% stained.

**Apoptosis**. Annexin-V/PI Apoptosis Detection Kit (BD Pharmingen) was used to assess cell apoptosis. Simply, Annexin-V and PI was used to label early and late apoptotic cells, respectively. Cells were suspended with binding buffer including Annexin-V for 15 min. Cells were washed and stained with PI in binding buffer. In order to conveniently set gates, cells were respectively stained with neither PI nor Annexin-V, PI alone, Annexin-V alone, and both all. After staining, cells were analyzed by FACScan flow cytometer (BD Biosciences).

**Magnetic resonance imaging**. After localizer imaging on three orthogonal axes, T1-weighted images of the entire mouse brain were acquired using a spin echo sequence with TR and TE set to 800 and 5.7 ms, respectively. Other parameters used were a 2.5 × 2.5 cm field of view and a 256 × 256 matrix in two averages, resulting in a total scan time of approximately 6.5 min. T2-weighted images were acquired with Repetition Time (TR) and Echo Time (TE) set to 4600 and 140 ms, respectively. Other parameters include 1 acquisition (NEX 2 averages), field of view = 6 × 6 cm, matrix = 256 × 256, slice thickness = 1 mm.

**Tumor size measurement**. Transduced U87-luc ($2 \times 10^5$) or TPC1115-luc ($2 \times 10^4$) cells were transplanted into the frontal lobes of brains of NOD-SCID IL-2 receptor gamma-null (NSG) mice by stereotaxic intracranial injection. Female BALB/C nude mice (4–6 weeks old) were purchased from Beijing Huafukang Bioscience Co., Inc. (Beijing, China) and used in all in vivo experiments. The PRMT2-H112Q or M115I expression in U87-luc cells were induced by feeding the xenografted NSG mice with drinking water containing 1 mg/ml Dox. Tumor growth was monitored by bioluminescence xenogen imaging. Briefly, the mice were anesthetized and given 150 μg/g of D-luciferin in PBS by intraperitoneal injection. The bioluminescence were then imaged with a charge-coupled device camera (IVIS; Xenogen). The survival of mice after cell transplantation was recorded and analyzed accordingly.

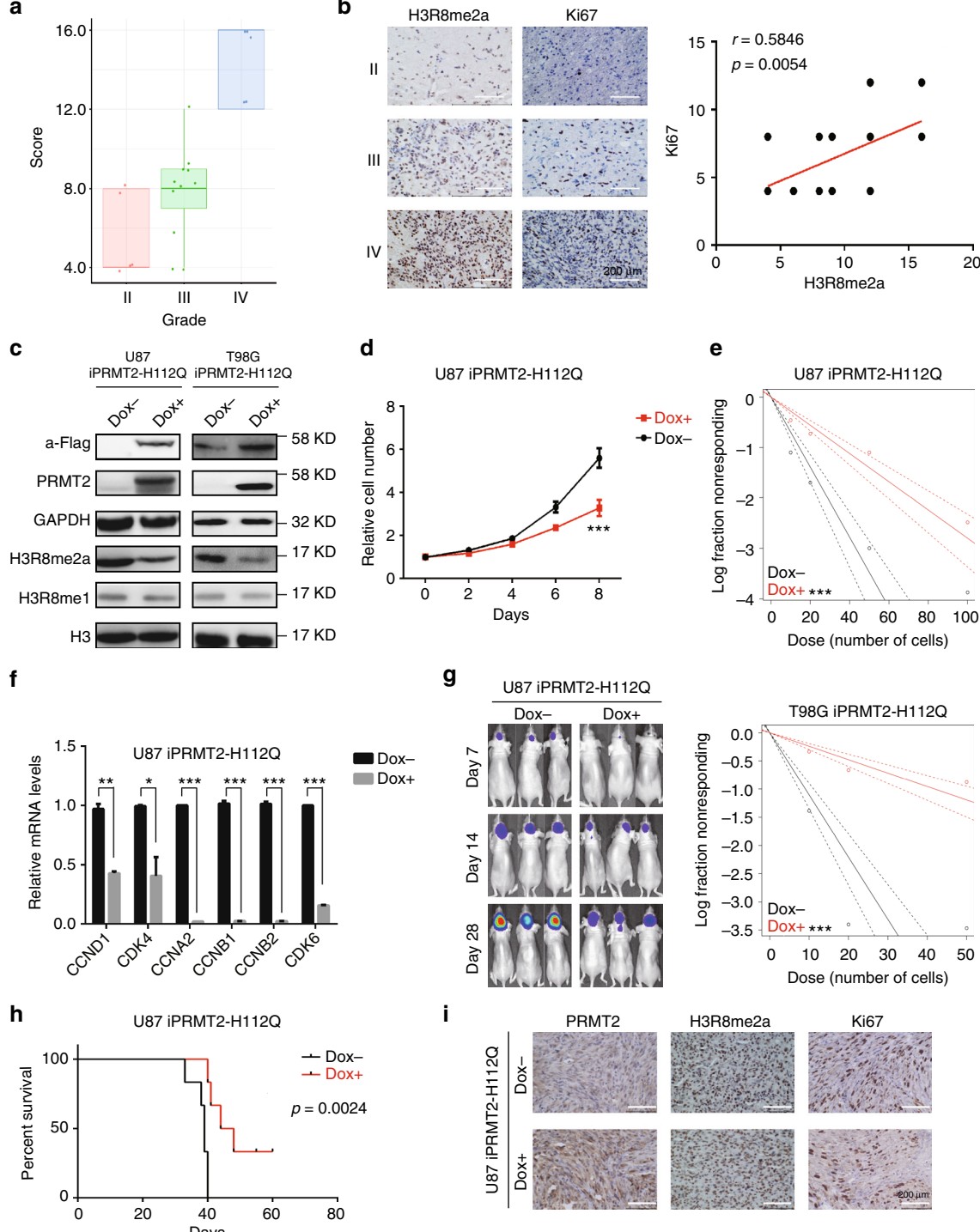

**RT-qPCR and immunoblotting**. Total RNA isolated using Trizol reagent (Ambion) was subjected to reverse transcription with Superscript Reverse Transcriptase (Invitrogen). RT-qPCR reactions were performed using SYBR Green Master Mix (Thermo) on 7500 Fast real-time PCR system (Applied Biosystems). *rPO* was used as a reference gene for all qRT-PCR experiments and analyses. The ΔΔCt method was used for quantification analysis. To obtain whole-cell protein extracts or histones, cells were lysed or extracted as previously described[46]. Membranes were blotted with the corresponding primary antibodies: anti-PRMT2 (1:1000, LifeSpan BioSciences), anti-STAT3 (1:1000, Abways), anti-p-STAT3 (1:1000, Abways), anti-α-Flag (1:2000, Sigma-Aldrich), anti-PTEN (1:1000, Cell Signaling Technology), anti-AXIN2 (1:500, Absin), anti-Histone H3 (1:4000, Millipore), anti-H3R8me2a (1:1000, Novus Biologicals), anti-H3K4me3 (1:1000, Cell Signaling Technology), anti-H3K27ac (1:1000, Abcam), and anti-GAPDH (1:5000, Abways). After careful washing, bound antibodies were detected with HRP-linked anti-mouse or anti-rabbit IgG (CST), followed by electrochemiluminescence (PerkinElmer) determination. All uncropped WBs can be found in Supplementary Figs. 11 and 12. Primer sequences and information for antibodies are available in Supplementary Table 2.

**RNA-seq and data analysis**. Construction of RNA-seq library and RNA-seq was completed by BGI company (Shenzhen, China). Single-end 1 × 50 bp sequencing were performed on BGISEQ-500. Raw data were processed using CASAVA V1.6 package. The quality control of each sample was accomplished using FASTQC V0.11.5. Clean reads were aligned to human reference genome hg19 by HISAT[47]. RSEM[48] was used to calculate gene expression, differential expression was determined by BGI's developed algorithm. Differential expressed genes (DEGs) with log2(fold change) > 1 and FDR < 0.001 were considered as significant. In order to unveil the pathways that may be associated with the identified DEGs, KEGG pathway enrichment analysis was performed using clusterProfiler package in R language[49].

**Fig. 7** PRMT2 methyltransferase activity is required for GBM cell growth and tumorigenesis. **a** Correlation between H3R8me2a levels and tumor grades. Tumor sections from 21 glioma specimens were IHC-stained with anti-H3R8me2a antibody. Lines within boxes indicate medians of the scores. **b** Correlation between H3R8me2a levels and Ki67 density in 21 GBM specimens. Representative examples of H3R8me2a and Ki67 immunostainings are shown in different grade of glioma specimens; scale bar, 200 μm (left panel). Stainings of nuclear H3R8me2a and Ki67 were scored and the significance of the correlation was determined by Pearson's correlation test (right panel). **c** Cellular levels of PRMT2 and H3R8me2a in U87 and T98G cells grown with or without PRMT2-H112Q expression. In order to induce PRMT2-H112Q expression, cells were treated with or without Doxycycline (Dox 1 mg/l) for 96 h. **d** Cell growth curves of U87 cells with or without PRMT2-H112Q expression. Significance level was determined using Student's two-sided $t$-tests. ***$p \leq$ 0.001. Error bars, ± SD, $n = 3$. **e** Frequency of sphere-initiating cells as measured by limiting dilution analysis in U87 (top panel) or T98G (bottom panel) cells with or without PRMT2-H112Q expression (mean ± SD, $n = 3$). Frequency and probability estimates were computed using the ELDA software. *$p \leq$ 0.05, **$p \leq$ 0.01, and ***$p \leq$ 0.001. **f** The relative mRNA levels of cell cycle associated genes were compared in U87 cells with or without PRMT2-H112Q expression. Significance level was determined using Student's two-sided $t$-tests. ***$p \leq$ 0.01. **g** Representative luciferase images of three mice per group at 7, 14, and 28 days post tumor implantation. **h** Survival analysis of mice intracranially implanted with U87 cells with or without PRMT2-H112Q expression. $X$ axis represents days after cells injection. Significance level was determined using log-rank analysis, $n = 6$ for each treatment group. *$p \leq$ 0.05. **i** Moribund mice were killed and dissected tumors were examined by IHC staining in two group specimens with or without PRMT2-H112Q expression. Scale bars, 200 μm

**Chromatin immunoprecipitation**. The regular chromatin preparation was performed as described[50]. In brief, cells were cross-linked with 1% formaldehyde solution for 10 min and quenched with 0.125 M glycine. For H3R8me2a ChIP, the cell pellets were resuspended in cold CSK buffer (100 mM NaCl, 300 mM sucrose, 3 mM $MgCl_2$, 10 mM PIPES, pH 6.8) containing Triton X-100 (0.5%) and EGTA (1 mM) before 10 min of fixation in 0.5% Formaldehyde solution. After being quenched by 0.125 M Glycine, cells were spun down, washed, resuspended, lysed, and ultra-sonicated for 25 min with 30 s ultra-sonication at 30 s intervals (Bioruptor pico, Diagenode, Belgium). The resulting fragmented chromatin extract was precleared with Protein A/G beads (ThermoFisher, Beijing, China) and then incubated overnight with antibodies: anti-H3K4me1, anti-H3K4me3, anti-3K27me3, anti-H3K9me3 (Cell Signaling Technology), anti-H3K27ac (Abcam), anti-H3R8me2a (Novus Biologicals), and Normal anti-rabbit IgG (Cell Signaling Technology) as controls. After stringent washes, elution, and reverse cross-linking, DNA was purified using PCR purification kits (QIAGEN, Hilden, Germany). The primers for ChIP-qPCR analyses at the promoters and enhancers are listed in Supplementary Table 2.

**ChIP-seq data processing**. Single-end $1 \times 50$ bp sequencing were performed on Illumina HiSeq 4000. The quality control of each sample was accomplished using FASTQC V0.11.5. Clean reads were aligned to human reference genome (hg19) using Bowtie 2 version 2.2.9[51]. SAMtools[52] was used to remove PCR duplicate and unmapped reads. Only unique mapped reads were used in further analysis. Peak calling and annotation were performed using MACS version 1.4.2[53] and ChIP-seeker package[54], respectively. Proximal regions around the TSS (TSS ± 3000 bp) were defined as promoters. Bigwig files, Heatmaps, and average profiles were generated by deepTools V2.4.0[55]. Hg19 annotated RefSeq genes list was downloaded from the UCSC database.

**Cohorts and transcriptome data**. To comprehensively explore the expression pattern and prognostic implications of PRMT family members in gliomas, whole transcriptome sequencing and corresponding clinical data (*IDH* mutation, WHO grades, transcriptome subtypes, and survival information) were downloaded from TCGA and CGGA (http://www.cgga.org.cn)[56]. The transcriptome data for TCGA and Repository for Molecular Brain Neoplasia Data (Rembrandt) were obtained from Gliovis (http://gliovis.bioinfo.cnio.es/) for analyses as well. One-way analysis of variance followed by Dunnett's multiple comparisons test and survival analyses were performed using GraphPad Prism version 6.00 for Windows.

**Gene-set enrichment analysis**. GSEA was performed with the public application from the Broad Institute[57]. Fragments per kilobase of transcript per million mapped reads values for all human genes generated from RNA-seq data were used for expression datasets. KEGG pathway gene sets and our defined PRMT2 positively correlated gene set were used for analysis. FDR was calculated by repeating sample permutations 1,000 times.

**Structure prediction**. The PDB model of human PRMT2 is built by SWISS-MODEL referring the model of *Mus musculus* PRMT2(PDB: 5FWD). The PDB of SAM is gained from human PRMT5 and SAM (PDB: $4 \times 61$). Furthermore, the binding model of human PRMT2 and SAM is predicted by using HADDOCK2.2 server that can input the possible active resides of binding SAM, when submitting the PDB of human PRMT2 and PAM, respectively. Finally, we can gain the binding model about human PRMT2 and SAM complex according to score value.

**Statistics**. No statistical method was used to predetermine sample size. The investigators were not blinded to allocation during experiments and outcome assessment. All grouped data are presented as mean ± SD. Unpaired Student's $t$-tests are presented as mean ± SD during comparison between unpaired two-groups and one-way ANOVA

was applied for multi-group data comparison. The variance was similar between the groups that were being statistically compared. Bivariate correlation analysis (Pearson's $r$ test) was used to examine the correlation of two variables in human specimens. All data met the assumptions of the tests.

Kaplan–Meier curves were generated and log-rank analysis was performed using GraphPad Software. Significant differences for all quantitative data were considered when *$p \leq$ 0.05, **$p \leq$ 0.01, ***$p \leq$ 0.001, and ****$p \leq$ 0.0001.

### Data availability

All relevant data supporting the key findings of this study are available within the article and its Supplementary Information files or from the corresponding author on reasonable request. RNA-seq and ChIP-seq data have been deposited into Gene Expression Omnibus (GEO) under accession GSE110424.

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

## Acknowledgements

We thank Fang Huang, Lei Lei for technical support. We are grateful to Tao Jiang for sharing CGGA transcriptome data. We thank Kristian Helin for providing comments on the manuscript. This work was supported by National key research and development program (2017YFA0504102 and 2016YFC0902502), the National Natural Science Foundation of China (81772676, 31570774, and 81702637), Tianjin Municipal Science and Technology Commission (17JCZDJC35200), and Talent Excellence Program from Tianjin Medical University.

## Author contributions

F.D. and X.Wu designed research. Q.L., Y.C. and X.L. performed bioinformatic analysis. F.D., D.H., C.Y., X.Wang, C.A., Y.K., X.S., W.Li., W.W. and W.G. performed research. Y.Z. and W.Liu contributed reagents. F.D., C.K. and X.W. analyzed data. F.D. and X.W. wrote the paper.

## Additional information

**Competing interests:** The authors declare no competing interests.

