## [Peer Review File · Nature Communications]

Reviewer #1 (Remarks to the Author):

This was a well-thought-out manuscript that identifies PRMT2 as a key epigenetic regulator of GBM progression. The authors provide convincing evidence to prove that PRMT2 is up-regulated in GBM, which confers a poor prognosis due to the transcriptional deregulation of key pathways that are important for the progression of cancer. The study is novel and PRMT2 role in GBM has not been described. The results add significant value to the biology of GBM. Below are some points that need to be addressed or clarified:

1. In the discussion, it was mentioned that there are chemical inhibitors targeting epigenetic modifiers. Is it feasible to use a non-specific methyltransferase inhibitor to potentially reverse the transcriptional effects mediated by PRMT2?
2. In Figure 7, a dox-inducible cell line was generated to induce the expression of a catalytic-mutant version of PRMT2. In the entire manuscript before this figure, PRMT2 expression was knocked down with siRNA. What happens if you overexpress PRMT2? Would enhanced proliferation be observed? Or would enhanced tumorigenesis be observed in animals? Would the gene signature be the exact opposite of the siRNA data? The data in Figure 7 would be strengthened with the overexpression of PRMT2 next the mutant data.
3. Why is the PRMT2-M115I catalytic mutant non-functional?
4. Fig 2f, it is difficult to read the figure legends and these should be enhanced.
5. The PTEN status of TPC1115 and TCO411 should be mentioned. Also, western blot for PRMT2 knockdown should be shown for these cells.
6. H&E for TPC115 should be included in Fig. 3g. Unlike U87 cells, the TIC cells should recapitulate the invasive nature of GBM. It would be valuable to assess whether the invasive phenotype of these cells are altered upon PRMT2 knockdown.
7. From the RNA seq data, how many genes are shared between PRMT2 knockdown in U87 and TPC115. Are the gene/pathway changes observed in U87 cells upon PRMT2 knockdown detected in TPC115 or vice versa? It would be helpful to correlate the genes affected by PRMT2 knockdown in Fig 4 d in the TCGA database. Do GBM specimens with overexpressed or low PRMT2 expression correlate with altered expression in CCNA2, CCNB1, CCNB2, etc?

Reviewer #2 (Remarks to the Author):

This study reports the role of PRMT2 in GBM. Authors show that PRMT2 expression level is correlated with glioma grade and predict GBM patient survival. Using shRNA to knockdown PRMT2 levels in established cell lines and primary tumorsphere cells, they report that PRMT2 is required for proliferation, sphere formation, cell cycle progression and GBM growth in vivo. They also show that PRMT2 KD results in specific downregulation of H3R8me2a, consistent with previous report by others that PRMT2 modulates H3R8me2a.

General comments:

- 1) Statistical analyses of all dataset presented must be more clearly indicated. Most panels are missing p-values and/or n-numbers despite statements in the manuscript that they show "significantly" changes. As is, it is unclear how consistent and reproducible key observations are. In addition, some analyses were performed with only a single shRNA in a single cell line (U87). Considering the increasing consensus in the field that U87 is not a good model of human GBM, authors should reproduce key observations in patient derived cells with multiple hairpins.
- 2) While the western blot shows dramatic downregulation of H3R8me2a mark in PRMT2 KD cells (Fig 5A), ChIP-seq peaks shown throughout the study do not appear significantly reduced. This discrepancy should be clearly explained as the main point of this report is the functional significance of PRMT2 in modulating gene expression through H3R8me2a control.
- 3) Authors conclude that gene expression and phenotypic changes observed with PRMT2 KD is due to PRMT2 regulation of H3R8me2a which also affects other histone marks associated with transcription activation. However they make an assumption that "regions with decreased H3R8me2a signals in PRMT2-depleted cells are PRMT2 target sites". However, with concerns expressed above and #4 below, it is not clear that effects on H3R8me2a is directly due to loss of PRMT2 alone. Since they claim that ChIP grade antibody for PRMT2 is not available, they should try a rescue experiment (in KD cells) with epitope-tagged PRMT2 expression and demonstrate that PRMT2 binds to the same regions and modulates H3R8me2a marks and gene expression.

Specific comments:

- 1) More detailed and controlled analyses of cell cycle progression are needed. There are no error bars and statistical analysis. Authors should also ensure that they plated the same number of VIABLE cells at log phase growth and follow their cell cycle progression over time since initial culture condition can have a large impact on cell cycle behavior. Also if PRMT2 KD depletes long term self-renewing cells as they show in Figure 3a, they should only use acute KD cells for analyses since stable KD cells have likely activated compensatory mechanisms to bypass PRMT2 KD. Passage numbers of experimental cells post shRNA transduction should be clearly indicated in each experiment.

- 2) Authors claim that there is no change in apoptosis but Supplementary figure 3 shows increased apoptosis with both hairpins. Authors should provide stronger evidence for their conclusion. They will need a rigorous statistical analysis to make a convincing conclusion either way.
- 3) Description for limiting dilution assay shown in Figure 3C does not match the figure legend. How many cells were really plated? They should also show estimated frequency of sphere forming cells and show statistical significance.
- 4) ChipSeq peaks shown in Figure 5F show differences between shScr and shPRMT2-1 but not necessarily overall decrease as authors claim. There appear to be peaks that are increased as well as decreased in KD cells. They need to demonstrate their conclusions more rigorously.
- 5) Author should be more careful in choosing accurate terms to describe their data. In vivo assays shown measure tumor growth, not "tumor progression". Similarly, PRMT2(H112Q) did not "abrogated" the in vivo tumor growth (Fig. 7g) since DOX+ brains clearly show growing tumors. Also in describing figure 7i, they should use "reduced" not "lost" since the panels clearly show H3R8me2a+ cells (or if there are no + cells, they need to show higher magnification image to demonstrate).
- 6) They should quantitate and perform statistical analysis to claim "significant reduction of tumor volume", shown in Figure 3E and 7g.
- 7) Authors should report how consistent the RNA-seq results were between U87 and TPC115 cells.
- 8) MRI scan mentioned in the text is not shown. Or is it Supplementary figure 4 (not listed in the text)?

Reviewer #3 (Remarks to the Author):

The Histone Code hypothesis –the mapping of a potential cross-talk between histones' marks– is at the forefront of Cancer Biology. The acetylation/methylation dynamic of histone lysine residues is well covered and its clear implications on genes' activation/repression are now well accepted concepts (H3K4 & K9). However, very little is known about the me1, me2a and me2s marks (-mono, asymmetric dimethyl and symmetric dimethyl, respectively) deposited by the protein arginine methyltransferases (PRMTs). The PRMTs are emerging therapeutic targets, so that new studies relevant to these enzymes will further support our understanding of such an intricate methyltransfer network. As this manuscript focuses onto the rather 'orphan' PRMT2 and its H3R8me2a deposited mark, the study reported herein is highly relevant and appropriate for this journal.

Focusing on glioblastoma multiform (GBM), the authors established the importance of PRMT2 (along with the deposition of H3R8me2a mark) and provide strong evidences to validate this enzyme as a novel therapeutic target. Highly expressed into the GBM system, the PRMT2 activity led to elevated and enriched levels of the me2a mark onto H3R8 residue at both promoter and enhancer regions. Furthermore, this mark correlated with other known Activation marks (i.e. H3K4me1 and H3K27Ac). To further investigate the role of PRMT2 and the impact of H3R8me2a onto gene regulation, the authors then depleted this methyltransferase through selective silencing of the enzyme. Through this process, with loss/decrease of the H3R8me2a mark, the authors deregulated the genes' expression pattern further supporting their hypothesis that the enzyme is linking H3R8me2a to oncogenic activation and tumorigenesis of the GBM system. As a final validation, the team focused onto the intrinsic enzymatic activity of PRMT2; a 'dead mutant' (H112Q) was generated. Overall, the results are in good agreements with the previous silencing of this target and support that H3R8me2a mark is mostly deposited by an active PRMT2 enzyme.

This manuscript is very well presented with both *in vitro* cell culture and *in vivo* tumor growth model to strongly support the findings. Furthermore, ChIP-seq allows for a thorough analysis. Pending the appropriate listed modifications, I will strongly recommend this manuscript for publication in *Nature Communications*.

Major

Regarding the PRMT2 enzymatic activity— The authors prepared two mutants (H112Q and M115I). This design was based upon a prediction of the SAM binding pocket of the enzyme.

The *Danio rerio* structure of PRMT2 was released earlier (11/09/16, PDB: 5FUB; FEBS J. 2017 284(1), 77-96 PMID:27879050). Through sequence and structural alignment, the human H112 and M115 residues are equivalent to H90 and M93, respectively. While histidine residue is crucial for cofactor binding (carboxylate tail of SAM; loss of activity observed in the current manuscript), a mutation of the M115 residue has more complex implications that are under investigated in this manuscript.

Indeed, previous mutation of rat PRMT1 M48 (M48A, M48L and M48F, PDB: 3Q7E; J Biol Chem. 2011 286(33), 29118-26), led to an alteration of product specificity (loss of the me2a deposition with an enzyme only able to catalyze the mono-methylation).

Although the authors focused onto the H3R8me2a mark, I wish to see complementary blots against the H3R8me1 mark. This short set of experiments would be:

An extra line within Fig. 5a for H3R8me1

An extra line within Fig. 7c for H3R8me1 (both U87 and T98G iPRMT2-H112Q)

An extra line within Sup Fig. 7b for H3R8me1 (U87 iPRMT2-M115I)

This will simply confirm that the me2a- but not me1-mark is indeed responsible for the observations.

Minor

Please be more rigorous with formatting and reference to figures and edit the following:

1) Within introduction, page 3 "...PRMT8) catalyze to form MMA and aSDMA..." to become "...PRMT8) catalyze to form MMA and ADMA...".

2) Same paragraph PRMT7 is either a Type II or Type III enzyme.

3) With section The arginine methyltransferase activity of PRMT2 is essential for its pro-tumorigenic functions, there is a reference to Sup Fig. 7b regarding cell proliferation. This is inaccurate and reference should be made to Sup Fig. 7c

4) Likewise, following reference to Sup Fig. 7c is inaccurate and is not pertinent to expression of regulated genes. A reference to panel d from Sup Fig. 7 should be used.

5) Sup Fig. 7a, if an X-ray structure was used, please provide origin organism and PDB number along with reference. If it is a model it must be specified how calculations were performed.

6) In Fig. 4, there are two panels labelled c.

7) In Fig. 5, there are two panels labelled b.

Thank you for giving us the opportunity to revise and resubmit this manuscript. We appreciate the time and details provided by each reviewer and have incorporated the suggested changes into the new version of the manuscript to the best of our ability. The new manuscript has certainly benefited from the reviewers' insightful suggestions. We hope that you will be able to accept the revised manuscript for publication in Nature Communications.

Point-by-point response to the referees

Referee 1:

We are pleased that the referee believes that our findings are interesting and we deeply appreciate the insightful comments. We have taken all the suggestions into serious consideration and we believe that the revision has greatly improved the quality of the manuscript.

Specific points and questions

1. In the discussion, it was mentioned that there are chemical inhibitors targeting epigenetic modifiers. Is it feasible to use a non-specific methyltransferase inhibitor to potentially reverse the transcriptional effects mediated by PRMT2?

Yes, dozens of chemical inhibitors targeting histone lysine methyltransferases or demethylases have been developed in the past decade. However the development of PRMT inhibitors are lagging far behind. Though several PRMT5 (type II) specific inhibitors have been tested, rather few selective inhibitors against other PRMTs are available. Nevertheless, we tried to do the experiments as the reviewer suggested. We treated GBM cells (U87 and TPC1115) with MS023, a wide-spectrum of Type I PRMT inhibitor, as PRMT2 belongs to the Type I of PRMTs. However MS023 inhibits the catalytic activity of PRMT1-mediated H4R3me2a, but leaves no effects on H3R8me2a (**Fig. 1 for Referees**). Thus we could not go further to examine the transcriptional effects of PRMT2 inhibitors at present. Currently we are collaborating with a chemical biologist on developing specific inhibitors targeting PRMT2.

We hope that we could have proper reagents to address this question. And it is also our ultimate goal to develop effective and specific small molecules targeting PRMT2 for preclinical trials of GBM treatment.

Fig. 1 for Referees WB analysis for the designated histone modifications in U87 and TPC1115 cells treated with MS023. H3 and GAPDH were used as loading controls.

2. In Figure 7, a dox-inducible cell line was generated to induce the expression of a catalytic-mutant version of PRMT2. In the entire manuscript before this figure, PRMT2 expression was knocked down with siRNA. What happens if you overexpress PRMT2? Would enhanced proliferation be observed? Or would enhanced tumorigenesis be observed in animals? Would the gene signature be the exact opposite of the siRNA data? The data in Figure 7 would be strengthened with the overexpression of PRMT2 next the mutant data. We thank the reviewer for the suggestion. We did ever try the overexpression of PRMT2 in GBM cells through regular expressing vectors and the Dox-inducible expression system. However no significant effects on cell proliferation were observed in either condition. It is understandable as epigenetic modifications or the modifiers are generally necessary but insufficient for cell fate decisions. In this study, we would like to only stress that PRMT2 is required for GBM development. Nevertheless, the reviewer makes an interesting point that we were planning to be the focus of a future study. To make clear whether PRMT2 acts as an oncogene, Prmt2 transgenic (WT and Mutant) mouse models will be generated. By crossing with other genetic tumorigenesis models, we would like to see whether the overexpression of WT-Prmt2 facilitates while the mutant attenuates the transformation process.

3. Why is the PRMT2-M115I catalytic mutant non-functional?

We thank the reviewer for the question but we are sorry that we do not have answers at the moment. Based on the published literature *Structural studies of protein arginine methyltransferase 2 reveal its interactions with potential substrates and inhibitors*, we predicted the pocket site for SAM docking on PRMT2 (Methods now included in **Supplementary Information**). Two sites, H112 and M115 were predicted to be important. However it turned out that H112 is more important for the catalytic activity of PRMT2 than M115 in our overexpression assay. Further structural studies are required to understand the details.

4. Fig 2f, it is difficult to read the figure legends and these should be enhanced.

We made a clearer description in the new figure legends as suggested and hope that it would be less confusing.

5. The PTEN status of TPC1115 and TC0411 should be mentioned. Also, western blot for PRMT2 knockdown should be shown for these cells.

We did the WB analysis for PTEN in two GBM cell lines and two TPC lines. The data is currently shown in **a new Fig. 1c and Supplementary Fig. 4a**. The significant depletion of PRMT2 at the protein levels in TPC cells were confirmed by WB analysis, now as shown in new **Supplementary Fig. 4b**.

6. H&E for TPC115 should be included in Fig. 3g. Unlike U87 cells, the TIC cells should recapitulate the invasive nature of GBM. It would be valuable to assess whether the invasive phenotype of these cells are altered upon PRMT2 knockdown.

Thank the reviewer for the suggestion. As for the invasive phenotype, the H&E staining for U87 orthotopic tumors reveals infiltration of neoplastic cells into the neighboring normal brain tissue and palisading necrosis in the margin. And this invasiveness is inhibited by PRMT2 depletion (Fig. 3g). Actually we also did the H&E staining for TPC1115 orthotopic tumors but did not show the data. As shown in **Fig. 2 for Referees**, it is probably more apparent to see the palisading effects around necrotic foci in TPC1115-derived tumors than in

U87 tumors. However as mentioned in Methods, we injected 2×10^4 TPC1115-luc cells in contrast to 2×10^5 U87-luc cells considering the stronger tumorigenic capacity of TPC cells. Because PRMT2 depletion in TPC1115 cells suppresses the tumorigenicity and cell proliferation, we failed to dissect any tumors in the PRMT2-KD group. Without the comparison, we would not like to include this data in the main figures. In the revised manuscript, we stress the importance of PRMT2 in GBM invasiveness in the description of Fig. 3g and in DISCUSSION.

Fig. 2 for Referees Dissected tumors of TPC1115 shScr groups were examined by IHC staining.

7. From the RNA seq data, how many genes are shared between PRMT2 knockdown in U87 and TPC1115. Are the gene/pathway changes observed in U87 cells upon PRMT2 knockdown detected in TPC1115 or vice versa? It would be helpful to correlate the genes affected by PRMT2 knockdown in Fig 4 d in the TCGA database. Do GBM specimens with overexpressed or low PRMT2 expression correlate with altered expression in CCNA2, CCNB1, CCNB2, etc?

As suggested, we did compare the overlapped deregulated genes in U87 and TPC1115 cells, though only around 20% of downregulated genes were shared, as shown in **new Supplementary Fig. 7b**. It is not surprising because the two cells were from distinct backgrounds and have been cultured in different conditions. Notably, the downregulated genes in either U87 or TPC1115 cells are similarly enriched in cell cycle progression and JAK/STAT3 signaling pathway (Fig. 4 and **new Supplementary Fig. 7c and d**).

We thank the reviewer for the insightful suggestion to establish the correlation of our DEGs with the TCGA dataset. After careful analyses, we have identified 583 genes whose

expression levels are positively correlated with PRMT2 expression in TCGA-GBM RNA-seq dataset, with p value < 0.001 and correlation coefficient > 0.3. Using these PRMT2-positively correlated genes as our created geneset, Gene Set Enrichment Analysis (GSEA) was performed. It turned out that PRMT2-positively correlated genes are significantly enriched in the control group (Scr), compared to PRMT2-depleted group (KD) (FDR 0.0045). The data was now shown in new Fig. 4d.

Furthermore we did the IHC staining of the clinical glioma samples in parallel for PRMT2 and the cell cycle genes with the IHC-grade antibodies available. As shown in new Supplementary Fig. 6, the PRMT2 expression levels are significantly correlated with the expression levels of CCND1, CCNB1 or CDK4 in different grades of gliomas. The significance of the correlation was determined by Pearson's correlation test. Therefore these bioinformatic analysis and IHC data confirm the transcriptional effects of PRMT2 as illustrated by our cell models.

Referee 2:

General comments:

1) Statistical analyses of all dataset presented must be more clearly indicated. Most panels are missing p-values and/or n-numbers despite statements in the manuscript that they show "significantly" changes. As is, it is unclear how consistent and reproducible key observations are. In addition, some analyses were performed with only a single shRNA in a single cell line (U87). Considering the increasing consensus in the field that U87 is not a good model of human GBM, authors should reproduce key observations in patient derived cells with multiple hairpins.

These critical comments are highly appreciated. In our revised version, we clearly marked the missing p-values according to proper statistics and n-numbers in new Figures. The statistics are also precisely described in the revised figure legends. And at least two shRNAs have been used to knock down PRMT2 in the patient-derived TPC1115 cells as well as in U87 cells for the *in vitro* or *in vivo* functional assays. We believe that these updated data and description will greatly strengthen this study.

2) While the western blot shows dramatic downregulation of H3R8me2a mark in PRMT2 KD cells (Fig 5A), ChiP-seq peaks shown throughout the study do not appear significantly reduced. This discrepancy should be clearly explained as the main point of this report is the functional significance of PRMT2 in modulating gene expression through H3R8me2a control. We thank the reviewer for the careful analysis of our data. But we disagree that there exists discrepancy between the ChIP-seq analysis with the WB data. As shown by the ChIP-seq analysis in Fig. 5d, the chromatin-bound H3R8me2a is as significantly reduced as WB (Fig. 5a). Notably, H3R8me2a is mainly distributed at distal intergenic and intragenic regions and correlated with known enhancer-associated histone marks (Fig. 5c and Fig. 6). Consistently, H3R8me2a peaks are wide peaks, more like H3K4me1 rather than H3K4me3, as viewed from the genomic snapshots. The peaks do not appear so significantly reduced in the PRMT2-knockdown cells because we over-zoomed in when taking the snapshots. We apologize for the improper presentation in the old version. To avoid misunderstanding, the zoomed-out view of the peaks are updated in our revised version. As shown in **new Fig. 5f and supplementary Fig 8**, the H3R8me2a enrichment is obviously downregulated on the enhancers and/or promoters of several representative target genes.

3) Authors conclude that gene expression and phenotypic changed observed with PRMT2 KD is due to PRMT2 regulation of H3R8me2a which also affects other histone marks associated with transcription activation. However they make an assumption that "regions with decreased H3R8me2a signals in PRMT2-depleted cells are PRMT2 target sites". Since they claim that ChIP grade antibody for PRMT2 is not available, they should try a rescue experiment (in KD cells) with epitope-tagged PRMT2 expression and demonstrate that PRMT2 binds to the same regions and modulates H3R8me2a marks and gene expression.

We thank the reviewer for the nice suggestions. Initially we tried the rescue experiments in PRMT2-KD cells with Flag-tagged PRMT2 exactly as suggested. However, as the PRMT2-knockdown cells (TPCs as well as U87) grew extremely slow, we could hardly have enough cells in response to the lentiviral transduction of Flag-tagged PRMT2. Thus we just took advantage of our inducible Flag-PRMT2 expression system and did Flag

ChIP-qPCR analysis. As shown in **new Supplementary Fig. 9**, the addition of Dox clearly induces the enrichment of Flag-PRMT2 at the enhancers and promoters of the target genes, but not at the *GAPDH* promoter. These data supports that the defined H3R8me2a-downregulated target sites are indeed bound by PRMT2.

Specific comments:

1) More detailed and controlled analyses of cell cycle progression are needed. There are no error bars and statistical analysis. Authors should also ensure that they plated the same number of VIABLE cells at log phase growth and follow their cell cycle progression over time since initial culture condition can have a large impact on cell cycle behavior. Also if PRMT2 KD depletes long term self-renewing cells as they show in Figure 3a, they should only use acute KD cells for analyses since stable KD cells have likely activated compensatory mechanisms to bypass PRMT2 KD. Passage numbers of experimental cells post shRNA transduction should be clearly indicated in each experiment.

We did careful statistical analysis for the percentages of the each cell phase between control and PRMT2-depleted U87 cells. And we did exactly as the reviewer's suggested to plate equal numbers of viable cells for the cell cycle analysis. The figures with updated p-value marks are now in **new Fig. 2f**.

As for the reviewer's concern about the stable knockdown cell after long passaging, we would like to make a clarification. As mentioned earlier, the PRMT2-depleted GBM cells almost stop growing every time after the shPRMT2 lentiviral transduction followed by selection. Thus all the KD cells harvested in this study have never been passaged at all. We have to freshly transduce a large amount of parental cells if necessary. Therefore, in response to the referee's comments, we do not believe the acute knockdown experiments with siRNAs will be more informative.

2) Authors claim that there is no change in apoptosis but Supplementary figure 3 shows increased apoptosis with both hairpins. Authors should provide stronger evidence for their conclusion. They will need a rigorous statistical analysis to make a convincing conclusion either way.

We thank the reviewer for the critical comments. A rigorous statistical analysis has been done

on the percentages of the apoptotic cells in each group. The p-values are above 0.1 compared with the control group in each of the PRMT2-knockdown group. Now the figures are updated as shown in new Supplementary Fig. 3b.

3) Description for limiting dilution assay shown in Figure 3C does not match the figure legend. How many cells were really plated? They should also show estimated frequency of sphere forming cells and show statistical significance.

We would like to thank the reviewer for the careful analysis of our data. Though we did limiting dilution assay in all the GBM cells, we presented the data with fixed number of cells (10 cells per well) in the old version of manuscript, which is less precise to calculate the frequencies of sphere formation. For the limiting dilution assays, we plated 100, 50, 20, 10 for U87 and the derived cells while plated 50, 20 and 10 cells for other cells because the sphere forming capabilities vary among the different cells. We have included the information in the **METHODS**. And the data are accordingly updated in new Fig. 3b,c and Fig. 7e. The estimated frequency of sphere forming cells are shown in Supplementary Fig 5a.

4) ChipSeq peaks shown in Figure 5F show differences between shScr and shPRMT2-1 but not necessarily overall decrease as authors claim. There appear to be peaks that are increased as well as decreased in KD cells. They need to demonstrate their conclusions more rigorously. As explained above, the improper presentation of the wide peaks caused misunderstanding. We sincerely apologize for it and update the snapshots with clearer differences in main figures and supplementary figures. We believe that the changes will avoid misunderstanding and greatly improve the quality of figures.

5) Author should be more careful in choosing accurate terms to describe their data. In vivo assays shown measure tumor growth, not "tumor progression". Similarly, PRMT2(H112Q) did not "abrogated" the in vivo tumor growth (Fig. 7g) since DOX+ brains clearly show growing tumors. Also in describing figure 7i, they should use "reduced" not "lost" since the panels clearly show H3R8me2a+ cells (or if there are no + cells, they need to show higher magnification image to demonstrate).

We deeply appreciate the reviewer's critical comments and made the necessary amendments in the revised manuscript.

6) They should quantitate and perform statistical analysis to claim "significant reduction of tumor volume", shown in Figure 3E and 7g.

The statistical analyses have been done according to the signals of luciferase activity in different groups of xenografted mice brain at different time points. The data are updated in new Fig. 3e and Supplementary Fig. 10e.

7) Authors should report how consistent the RNA-seq results were between U87 and TPC115 cells.

As mentioned in the response to Reviewer 1, we did compare the overlapped deregulated genes in U87 and TPC115 cells. As shown in new Supplementary Fig. 7b, around 20% of downregulated genes were shared in the two different cell models. This is not surprising considering about the heterogeneity of GBM and different culture conditions. As we know, even not so many deregulated genes are shared in patients-derived TPCs (Figure 3A in Ref: Zhang, S., et al. (2017). "m6A Demethylase ALKBH5 Maintains Tumorigenicity of Glioblastoma Stem-like Cells by Sustaining FOXM1 Expression and Cell Proliferation Program." *Cancer Cell* 31(4): 591-606 e596.). Nevertheless, the downregulated genes in the two cell models are similarly enriched in cell cycle progression and JAK/STAT3 signaling pathway, no matter for the overlapped or un-overlapped.

8) MRI scan mentioned in the text is not shown. Or is it Supplementary figure 4 (not listed in the text)?

Yes, the MRI scan was the old supplementary Fig 4 and already mentioned in the text. The reviewer probably missed it. Now it is presented and listed as new Supplementary Fig 5c.

Referee 3:

We would like to thank this reviewer for the very positive feedback on our study and the

strong recommendation to the Nature Communications readership.

Major Comments:

Regarding the PRMT2 enzymatic activity— The authors prepared two mutants (H112Q and M115I). This design was based upon a prediction of the SAM binding pocket of the enzyme. The *Danio rerio* structure of PRMT2 was released earlier (11/09/16, PDB: 5FUB; FEBS J. 2017 284(1), 77-96 PMID:27879050). Through sequence and structural alignment, the human H112 and M115 residues are equivalent to H90 and M93, respectively. While histidine residue is crucial for cofactor binding (carboxylate tail of SAM; loss of activity observed in the current manuscript), a mutation of the M115 residue has more complex implications that are under investigated in this manuscript. Indeed, previous mutation of rat PRMT1 M48 (M48A, M48L and M48F, PDB: 3Q7E; J Biol Chem. 2011 286(33), 29118-26), led to an alteration of product specificity (loss of the me2a deposition with an enzyme only able to catalyze the mono-methylation). Although the authors focused onto the H3R8me2a mark, I wish to see complementary blots against the H3R8me1 mark. This short set of experiments would be:

An extra line within Fig. 5a for H3R8me1

An extra line within Fig. 7c for H3R8me1 (both U87 and T98G iPRMT2-H112Q)

An extra line within Sup Fig. 7b for H3R8me1 (U87 iPRMT2-M115I)

This will simply confirm that the me2a- but not me1-mark is indeed responsible for the observations.

We thank the review for the insightful suggestions. And we did H3R8me1 WB analysis as suggested. As shown in the **updated Fig. 5a, 7c and Supplementary Fig. 10b**, we did not find any obvious changes of H3R8me1 in the PRMT2 knockdown or mutant cells. It indicates that the deregulation of H3R8me2a is indeed responsible for the transcriptional effects and therefore the functional outcomes.

Minor Comments:

Please be more rigorous with formatting and reference to figures and edit the following:

1) Within introduction, page 3 "...PRMT8) catalyze to form MMA and aSDMA..." to become

“...PRMT8) catalyze to form MMA and ADMA...”.

2) Same paragraph PRMT7 is either a Type II or Type III enzyme.

3) With section The arginine methyltransferase activity of PRMT2 is essential for its pro-tumorigenic functions, there is a reference to Sup Fig. 7b regarding cell proliferation. This is inaccurate and reference should be made to Sup Fig. 7c

4) Likewise, following reference to Sup Fig. 7c is inaccurate and is not pertinent to expression of regulated genes. A reference to panel d from Sup Fig. 7 should be used.

6) In Fig. 4, there are two panels labelled c.

7) In Fig. 5, there are two panels labelled b.

We deeply appreciate that the reviewer read our manuscript so carefully and apologize for the previous carelessness. Now they are all corrected in the revised manuscript.

5) Sup Fig. 7a, if an X-ray structure was used, please provide origin organism and PDB number along with reference. If it is a model it must be specified how calculations were performed.

To make it clear, the structure data is just a model. Based on the published literature *Structural studies of protein arginine methyltransferase 2 reveal its interactions with potential substrates and inhibitors*, the PDB model of human PRMT2 was built by SWISS-MODEL referring the model of *Mus musculus* PRMT2 (PDB: 5FWD). The PDB of SAM was gained from human PRMT5 and SAM (PDB: 4X61). Then the active residues of human PRMT2 binding to SAM were predicted according to the scores from the HADDOCK2.2 server. The detailed description has been included in the Supplementary Information.

Reviewer #1 (Remarks to the Author):

The authors addressed all my concerns. The study is strong and the manuscript is improved.

Reviewer #2 (Remarks to the Author):

This is much improved manuscript and most of my concerns were addressed.

However, I do suggest that authors review /revise the cell cycle analyses.

It looks like the effect of PRMT2 KD on apoptosis and cell cycle arrest are similar- there are general trends towards increased apoptosis and G1 arrest but neither are statistically significant. Changes to G2 populations in the two cell lines are statistically significant but OPPOSITE in directions. All together, these data show equivocal or inconsistent effects on cell cycle progression. Therefore, text (lines 138-140) should be revised (and move Figure 2f to supplementary figure), or authors should repeat the experiments (currently n=2) to show statistically significant effects before concluding that “PRMT2 is required for cell cycle progression” in general.

Reviewer #3 (Remarks to the Author):

I thank the authors for making the suggested edits and additional experiments regarding their manuscript “PRMT2 links histone H3R8 asymmetric dimethylation to oncogenic activation and tumorigenesis of glioblastoma”. The manuscript is now suitable for publication in Nature Communications.

Dear Editors,

Thank you for giving us the opportunity to revise this manuscript again. We deeply appreciate you and the reviewers' efforts into improving the quality of our work. According to the suggestion of Reviewer #2, we have made the corrections. And the whole manuscript is updated according to the Editorial Request. Please feel free to let us know if there is still anything unsuitable for publication in Nature Communications.

Response to the referees

Referee 2:

This is much improved manuscript and most of my concerns were addressed. However, I do suggest that authors review /revise the cell cycle analyses. It looks like the effect of PRMT2 KD on apoptosis and cell cycle arrest are similar- there are general trends towards increased apoptosis and G1 arrest but neither are statistically significant. Changes to G2 populations in the two cell lines are statistically significant but OPPOSITE in directions. All together, these data show equivocal or inconsistent effects on cell cycle progression. Therefore, text (lines 138-140) should be revised (and move Figure 2f to supplementary figure), or authors should repeat the experiments (currently n=2) to show statistically significant effects before concluding that "PRMT2 is required for cell cycle progression" in general.

We are pleased that our efforts have addressed the reviewer's concerns. Meanwhile we sincerely thank the reviewer for the careful reading of our manuscript and pointing out the confusing data of our statistics in the cell cycle analysis. We apologize that we mislabelled asterisks on G1 phase in our original manuscript. And the experiments were performed twice (n=2) with the two shRNA transduced cell lines that have been *in vitro* cultured for different time. To avoid the inconsistency of repeats or inconclusion, we repeated the experiments with the two shRNA freshly transduced cell lines in triplicate. As shown in the new Fig. 2f, G1 arrest is statistically significant in both cell lines as previously found out, and G2 arrest is only significant in T98G cells. The difference may lie in the distinct genetic backgrounds, for example *PTEN* wild type in T98G in contrast to *PTEN* loss in U87 cells. Thus the updated data support our previous conclusion.